# Baicalein Resensitizes Multidrug-Resistant Gram-Negative Pathogens to Doxycycline

Yuhang Wang,[a,b] Junfeng Su,[a] Ziyan Zhou,[b] Jie Yang,[a,c] Wenjuan Liu,[d] Yafen Zhang,[d] Pengyu Zhang,[a,b] Tingting Guo,[a,b,c] Guocai Li[a,b,c,d,*]

[a]Department of Microbiology, Institute of Translational Medicine, Medical College, Yangzhou University, Yangzhou, People's Republic of China
[b]Jiangsu Key Laboratory of Experimental & Translational Non-coding RNA Research, Yangzhou, People's Republic of China
[c]Jiangsu Key Laboratory of Zoonosis/Jiangsu Co-Innovation Center for Prevention and Control of Important Animal Infectious Diseases and Zoonoses, Yangzhou University, Yangzhou, People's Republic of China
[d]Laboratory Department, Affiliated Hospital of Yangzhou University, Yangzhou, People's Republic of China

**ABSTRACT**  As multidrug-resistant pathogens emerge and spread rapidly, novel antibiotics urgently need to be discovered. With a dwindling antibiotic pipeline, antibiotic adjuvants might be used to revitalize existing antibiotics. In recent decades, traditional Chinese medicine has occupied an essential position in adjuvants of antibiotics. This study found that baicalein potentiates doxycycline against multidrug-resistant Gram-negative pathogens. Mechanism studies have shown that baicalein causes membrane disruption by attaching to phospholipids on the Gram-negative bacterial cytoplasmic membrane and lipopolysaccharides on the outer membrane. This process facilitates the entry of doxycycline into bacteria. Through collaborative strategies, baicalein can also increase the production of reactive oxygen species and inhibit the activities of multidrug efflux pumps and biofilm formation to potentiate antibiotic efficacy. Additionally, baicalein attenuates the lipopolysaccharide-induced inflammatory response *in vitro*. Finally, baicalein can significantly improve doxycycline efficacy in mouse lung infection models. The present study showed that baicalein might be considered a lead compound, and it should be further optimized and developed as an adjuvant that helps combat antibiotic resistance.

**IMPORTANCE**  Doxycycline is an important broad-spectrum tetracycline antibiotic used for treating multiple human infections, but its resistance rates are recently rising globally. Thus, new agents capable of boosting the effectiveness of doxycycline need to be discovered. In this study, it was found that baicalein potentiates doxycycline against multidrug-resistant Gram-negative pathogens *in vitro* and *in vivo*. Due to its low cytotoxicity and resistance, the combination of baicalein and doxycycline provides a valuable clinical reference for selecting more effective therapeutic strategies for treating infections caused by multidrug-resistant Gram-negative clinical isolates.

**KEYWORDS**  baicalein, multidrug-resistant, Gram-negative pathogens, doxycycline

Address correspondence to Guocai Li, gcli@yzu.edu.cn.

*Present address: Guocai Li, Medical College of Yangzhou University, Yangzhou, China.

The authors declare no conflict of interest.

Antibiotics are an important part of modern medicine and constitute the primary method of treating bacterial infections. However, the development of multidrug resistance (MDR) in patients complicates antimicrobial therapy, particularly for Gram-negative pathogens that have highly impermeable outer membranes; thus, MDR causes a clinical dilemma for clinicians (1). Globally, carbapenem-resistant Gram-negative opportunistic pathogens are rated as the top priority, along with *Pseudomonas aeruginosa*, *Klebsiella pneumoniae*, and *Acinetobacter baumannii* (2).

Gram-negative opportunistic pathogens develop various defense mechanisms for resisting the hazardous effect of antibiotics. The limited efficacy of antibiotics is due to the presence of MDR genes and intrinsic tolerance mechanisms, such as drug efflux (3),

biofilm formation (4), a unique impermeable outer membrane barrier (5), and other phenotypic resistance mechanisms. This phenotypic resistance is often associated with the tricarboxylic acid (TCA) cycle (6), proton motive force (PMF) (7), bacterial respiration, and reactive oxygen species (ROS) (8). For mitigating antibiotic resistance, a suitable alternative to the search for new antibiotics is the development of innovative strategies to increase the potency of known antibiotics.

Doxycycline is a broad-spectrum antibiotic targeting bacterial ribosomes that belongs to the tetracycline class. It is extensively used in clinics (9). However, with the indiscriminate use of antibiotics, the prevalence of doxycycline-resistant pathogens threatens public safety (10). The *tetA* efflux pump family is closely associated with doxycycline resistance (11). Therefore, developing practical approaches to repress *tetA* and combat doxycycline resistance is essential. Several studies have shown that traditional Chinese medicine combined with antibiotics is effective against bacterial infections (12–14).

Baicalein is a bioactive flavonoid extracted from the roots of *Scutellaria baicalensis* Georgi. It has been used to treat bacterial infections since ancient times (15). Jang et al. reported that baicalein inhibits *tetK* and prevents tetracycline from effluxing, thus resensitizing *Escherichia coli* to tetracycline (16). The methicillin-resistant *Staphylococcus aureus* strains OM481 and OM584 were suppressed synergistically by baicalein and tetracycline (17). Baicalein also has a synergistic effect when combined with cefotaxime on certain *K. pneumoniae* strains expressing *CTX-M-1* mRNA (18). Baicalein with gentamicin has a synergistic effect on vancomycin-resistant *Enterococcus* isolates (19). However, the mechanism through which baicalein acts as a broad-spectrum antibacterial adjuvant is not fully known.

In this study, we analyzed the broad-spectrum antibacterial adjuvant baicalein and the synergistic effect of baicalein combined with doxycycline. Detailed studies showed that baicalein could attach to phospholipids, compete with $Mg^{2+}$ to bind to lipid A, and exert antimicrobial effects on Gram-negative bacteria. Additionally, baicalein accelerated the TCA cycle, enhanced oxidative damage and bacterial cell membrane potential ($\Delta\Psi$) depolarization, leading to inhibition of multidrug efflux systems, and biofilm formation, thus improving the efficacy of doxycycline. Baicalein also down-modulated the lipopolysaccharide-induced host inflammatory response. These findings suggested that baicalein combined with doxycycline is efficacious in treating disorders resulting from MDR Gram-negative pathogens. Our study provides novel insights into treatment strategies to combat such infections.

## RESULTS

**Baicalein is a potent broad-spectrum antibiotic adjuvant.** For our previously constructed extensively drug-resistant (XDR) strain *A. baumannii* AB43Δ*crispr-cas* (20), a checkerboard assay showed that baicalein can act as a synergistic antibacterial activity (fractional inhibitory concentration index [FICI] ≤ 0.5) with most conventional antibiotics (10/14, 71.43%), including doxycycline (FICI = 0.1875) (Table 1). A similar pattern was found for the sensitive strain AB43 (Table S1). The combination of baicalein and doxycycline was effective against 245 clinical isolates of *A. baumannii* along with other *tetA*-positive MDR and pandrug-resistant (PDR) strains of Gram-negative pathogens (Fig. 1; Table 2). Our time-dependent killing curve showed that the combination treatment had good antibacterial activity against the *tetA*-positive MDR strains *A. baumannii* AB145 (FICI = 0.09375) and *K. pneumoniae* KP1 (FICI = 0.1875) (Fig. 2A and B). Relative to doxycycline alone, 62.5 $\mu$g/mL baicalein significantly enhanced the antimicrobial activity of doxycycline and reduced the number of viable AB145 and KP1 bacteria after 16 h of treatment *in vitro* (Fig. 2C). The results of the live/dead bacterial staining assay showed that the cell survival rate in the combined drug group was lower than that in the single-agent group (Fig. 2D and E).

Whether the combination therapy can exacerbate single-agent toxicities needs to be determined. Further examination of the biological safety of this synergism showed that even at high concentrations, the combination of baicalein and doxycycline caused

**TABLE 1** Synergistic activity of baicalein in combination with different classes of antibiotics against extensively drug-resistant AB43Δ*crispr-cas*

| Antibiotic | MIC (μg/mL) | FICI[a] | MIC with baicalein (μg/mL)[b] | Potentiation (fold) |
|---|---|---|---|---|
| Ampicillin | 10,192 | 0.28125 | 637 | 16 |
| Doxycycline | 128 | 0.1875 | 8 | 16 |
| Erythromycin | 32 | 0.375 | 2 | 16 |
| Rifampin | 8 | 0.25 | 1 | 8 |
| Minocycline | 4 | 0.3125 | 0.0625 | 64 |
| Tetracycline | 2,048 | 0.375 | 512 | 4 |
| Tigecycline | 64 | 0.375 | 8 | 8 |
| Imipenem | 16 | 0.25 | 4 | 4 |
| Gentamicin | 256 | 0.5 | 64 | 4 |
| Cefoperazone and sulbactam | 128 | 0.25 | 16 | 8 |
| Ceftriaxone sodium | 256 | 0.75 | 128 | 2 |
| Ciprofloxacin | 16 | >1 | ≥16 | ≤1 |
| Oxytetracycline | 32 | 1 | 16 | 2 |
| Vancomycin | 32 | 1.5 | 32 | 1 |

[a]FICIs were calculated based on checkerboard broth microdilution assays. Effects are defined as follows: synergy, FICI ≤ 0.5; additive, 0.5 < FICI < 1; indifferent, 1 ≤ FICI < 4.
[b]MIC in the presence of 62.5 μg/mL of baicalein.

only 1% hemolysis of sheep red blood cells (see Fig. S1A in the supplemental material). The mammalian cell cytotoxicity results consistently showed that the synergism of baicalein and doxycycline had low cytotoxicity (Fig. S1B to D) (21). Antibiotic resistance is a significant challenge in the treatment of bacterial infections. Serial passaging in the presence of doxycycline for 31 days increased MICs for *A. baumannii* AB145 and *K. pneumoniae* KP1 by 32- and 64-fold, respectively, whereas the combination treatment prevented the development of resistance (Fig. 2F). It is worth noting that the per-generation doxycycline-resistant mutants exhibited cross-resistance (22) to different classes of antibiotics, and the expression levels of *tetA* increased significantly in some high-level doxycycline-resistant mutants (Fig. S2; Table S2). Moreover, all doxycycline-resistant mutants contained multiple nonsynonymous mutations in the genes of *tetA* (Table S3). These results indicated that baicalein could inhibit doxycycline resistance in *tetA*-positive MDR *A. baumannii* and *K. pneumoniae* KP1.

**Baicalein disrupts the inner and outer membranes of Gram-negative bacteria.** Baicalein showed synergistic effects with multiple antibiotics against Gram-negative pathogens. We inferred that baicalein affects the endogenous mechanism of resistance to antibiotics in Gram-negative bacteria. The cell envelope of Gram-negative bacteria is the first line of defense against antibiotics (23). In this study, the fluorescence intensity of propidium iodide (PI) and 1-*N*-phenylnaphthylamine (NPN) and the extracellular

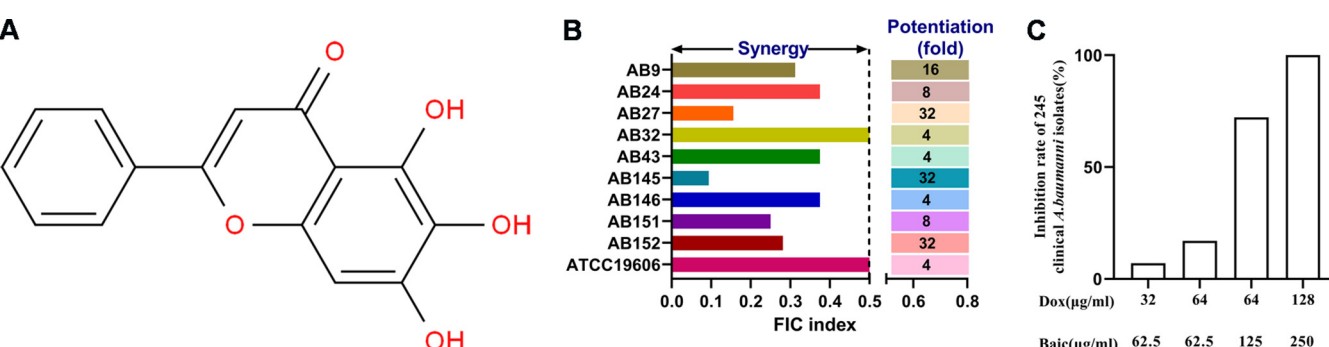

**FIG 1** Baicalein combined with doxycycline was effective against *A. baumannii*. (A) Chemical structure of baicalein. (B) Using a checkerboard microdilution assay, the synergy between baicalein and doxycycline against 10 representative clinical isolates of *A. baumannii* was determined. The degree of doxycycline potentiation in the presence of 0.25× MIC of baicalein was evaluated. Synergy was defined as a FICI of ≤0.5. (C) Combinations of baicalein and doxycycline against 245 clinical isolates of *A. baumannii*. According to the CLSI 2021 guidelines, when bacteria coincubated with baicalein and doxycycline for 18 h at 37°C exhibited no visible growth, we considered that growth inhibition occurred. Dox, doxycycline; Baic, baicalein. The data are based on three biological replicates.

**TABLE 2** Baicalein enhances the antimicrobial activity of doxycycline against Gram-negative pathogens

| Strain | Genotype[a] | Resistance pattern | Doxycycline | | Baicalein | | FICI[b] |
|---|---|---|---|---|---|---|---|
| | | | MIC ($\mu$g/mL) | FIC | MIC ($\mu$g/mL) | FIC | |
| *A. baumannii* | | | | | | | |
| AB9 | tetA$^+$ tetB$^+$ | MDR | 128 | 0.0625 | 250 | 0.25 | 0.3125 |
| AB24 | tetB$^+$ | PDR | 128 | 0.125 | 500 | 0.25 | 0.375 |
| AB27 | tetA$^+$ tetB$^+$ | MDR | 128 | 0.03125 | 250 | 0.125 | 0.15625 |
| AB32 | tetA$^+$ tetB$^+$ | PDR | 128 | 0.25 | 250 | 0.25 | 0.5 |
| AB43 | tetA tetB | Sensitive | 1 | 0.25 | 250 | 0.125 | 0.375 |
| AB145 | tetA$^+$ | MDR | 128 | 0.03125 | 250 | 0.0625 | 0.09375 |
| AB146 | tetB$^+$ | PDR | 128 | 0.25 | 250 | 0.125 | 0.375 |
| AB151 | tetA tetB | PDR | 64 | 0.125 | 500 | 0.125 | 0.25 |
| AB152 | tetA$^+$ tetB$^+$ | MDR | 64 | 0.03125 | 500 | 0.25 | 0.28125 |
| ATCC 19606 | tetA tetB | Type strain, non-MDR | 1 | 0.25 | 250 | 0.25 | 0.5 |
| *K. pneumoniae* | | | | | | | |
| KP1 | tetA$^+$ | MDR | 128 | 0.125 | 250 | 0.0625 | 0.1875 |
| KP2 | tetA$^+$ | PDR | 128 | 0.25 | 500 | 0.125 | 0.375 |
| KP6 | tetA$^+$ | PDR | 64 | 0.125 | 500 | 0.25 | 0.375 |
| KP10 | tetA$^+$ | MDR | 64 | 0.0625 | 125 | 0.125 | 0.1875 |
| KP12 | tetA$^+$ | MDR | 64 | 0.03125 | 250 | 0.25 | 0.28125 |
| KP13 | tetA$^+$ | PDR | 64 | 0.0625 | 500 | 0.25 | 0.3125 |
| KP14 | tetA$^+$ | PDR | 32 | 0.125 | 500 | 0.25 | 0.375 |
| KP15 | tetA$^+$ | MDR | 32 | 0.125 | 250 | 0.125 | 0.25 |
| *E. coli* | | | | | | | |
| E.col-1 | tetA$^+$ | MDR | 32 | 0.125 | 250 | 0.25 | 0.375 |
| E.col-2 | tetA$^+$ | MDR | 64 | 0.0625 | 500 | 0.0625 | 0.125 |
| E.col-3 | tetA$^+$ | MDR | 32 | 0.125 | 250 | 0.125 | 0.25 |
| *P. aeruginosa* | | | | | | | |
| PA-1 | tetA$^+$ | MDR | 64 | 0.125 | 250 | 0.125 | 0.25 |
| PA-2 | tetA$^+$ | MDR | 128 | 0.125 | 500 | 0.125 | 0.25 |

[a]+, presence; no symbol, absence.
[b]The FICI is the sum of the FICs of doxycycline and baicalein. All effects shown here are defined as synergistic (FICI $\leq$ 0.5).

content of $\beta$-galactosidases and ATP increased with the concentration of baicalein (Fig. 3 and Fig. S3). The scanning electron microscopy (SEM) images revealed that after the combination treatment, the cell surface was depressed and the cell shrank, collapsed, and underwent lysis, but these changes did not occur in untreated cells. This indicated that the effect of baicalein on bacteria might be associated with the rapidly disrupted cell wall and the inner and outer membranes of the cell.

In Gram-negative bacteria, lipopolysaccharide (LPS) is a significant component of the outer membrane. The exogenous addition of LPS from the corresponding bacterial strain reduced the antibacterial activity of baicalin in a dose-dependent manner, and high concentrations of LPS (128 $\mu$g/mL) reduced the antibacterial activity of baicalein combined with doxycycline (Fig. 4A and B). Through ionic bridges, divalent cations can connect to the negatively charged phosphate groups among lipid A molecules, which along with core polysaccharides can form LPS. We found that the exogenous addition of $K^+$, $Ca^{2+}$, $Mg^{2+}$, $Cu^{2+}$, and $Zn^{2+}$ inhibited the activity of baicalein, and $Mg^{2+}$ strongly inhibited the antibacterial activity of baicalein combined with doxycycline, while $Fe^{3+}$, $Mn^{2+}$, and $Na^+$ had no significant impact on these activities (Fig. 4C to E). As expected, the expression of *mgtA*, a transcriptional regulator responsible for $Mg^{2+}$ starvation, showed a dose-dependent increase with baicalein treatment, but the opposite occurred with *tetA* (Fig. 4F and G). The main functional and structural components of the inner membrane of Gram-negative bacteria are phospholipids, which include phosphatidylethanolamine (PE), cardiolipin (CL), and phosphatidylglycerol (PG) (24). The MICs of baicalein were higher with the exogenous addition of phospholipids, and PG blocked the antibacterial activity of baicalein combined with antibiotics (Fig. 5). These findings suggested that baicalein-mediated membrane perturbation strongly promotes the antibacterial activity of doxycycline.

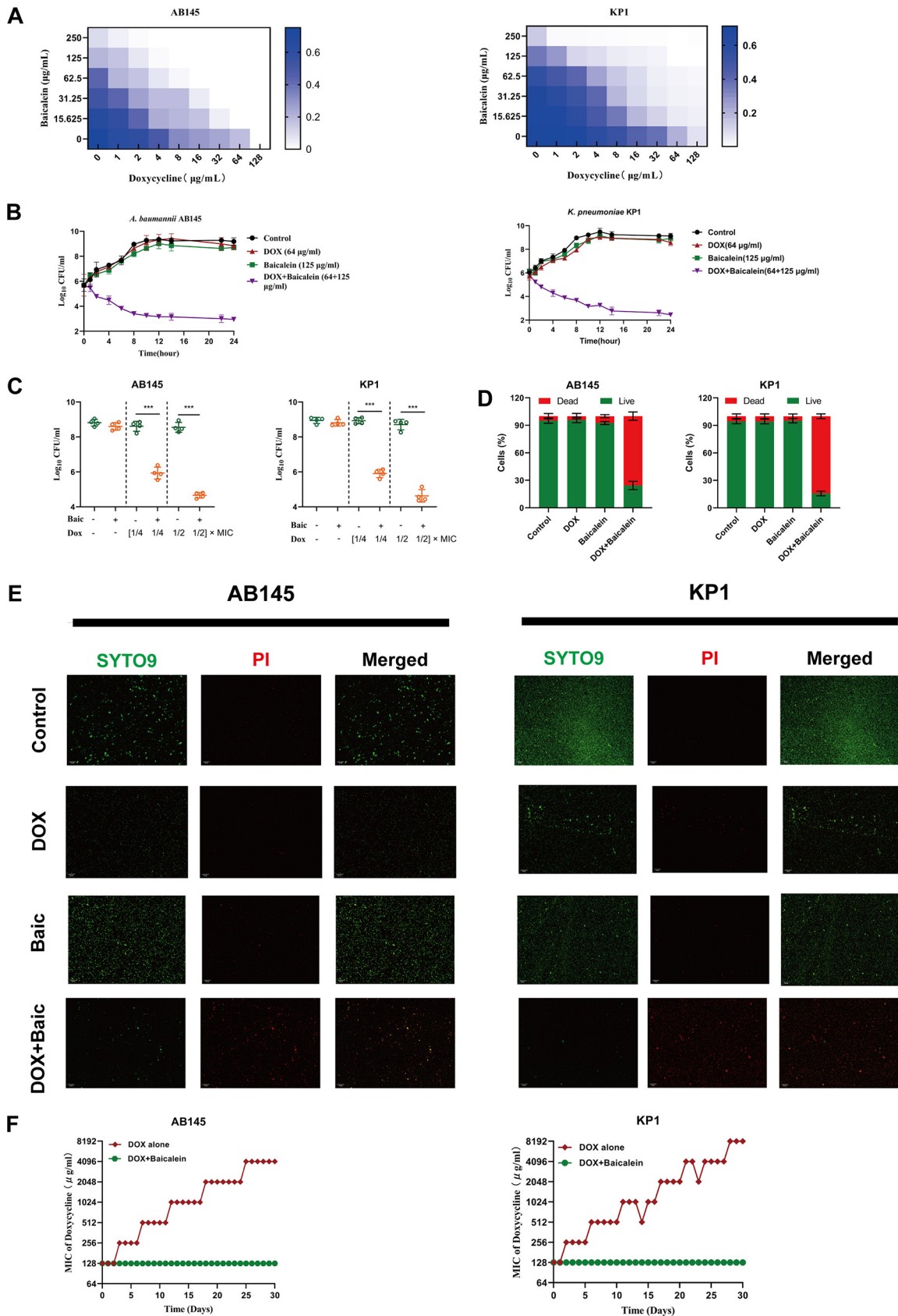

**FIG 2** Baicalein synergy with doxycycline has antibacterial activity against *tetA*-positive Gram-negative pathogens *in vitro*. (A) Synergistic antimicrobial effects of baicalein and doxycycline on *A. baumannii* AB145 and *K. pneumoniae* KP1, determined by the

**Transcriptomic analysis of the antibacterial activity of baicalein combined with doxycycline.** To better understand the inhibitory effect of baicalein combined with doxycycline in Gram-negative bacteria, transcription analysis of *A. baumannii* AB145 was conducted following the administration of doxycycline, baicalein, or their combination for 4 h. Compared to the treatment with doxycycline alone, the combination treatment upregulated 552 genes and downregulated 313 genes, while the combination group showed 704 upregulated genes and 505 downregulated genes compared to the group treated with baicalein alone (Fig. 6A). The results of KEGG analysis showed that the common differentially expressed genes were associated with microbial metabolism, protein export, oxidative phosphorylation, TCA cycle, and ABC transporters (Fig. 6B). Additionally, genes related to ATP synthase, NADH-quinone oxidoreductase, the multidrug efflux pump, ABC transporters, and membrane proteins were downregulated, whereas ROS-related genes were significantly upregulated after combination treatment compared to after treatment with doxycycline alone (Fig. 6C). Thus, the antibacterial activity of baicalein combined with doxycycline might be associated with multiple pathways.

**Mechanisms by which baicalein potentiates doxycycline activity.** According to the transcriptomic results, the NADH genes necessary for the TCA cycle were downregulated after combination treatment compared to after treatment with doxycycline alone. In agreement with this result, the supplementation of baicalein decreased the $NAD^+$/NADH ratio, which indicated that the TCA cycle activity was enhanced when combination treatment was administered (Fig. 7A; Fig. S4A and S5A). Enhanced TCA cycle activity in bacteria is usually accompanied by ROS generation (25). It is immediately followed by increased intracellular ROS levels and reduced superoxide dismutase (SOD) levels in combined treatment (Fig. 7B; Fig. S4B and S5). These results indicated that baicalein combined with doxycycline can increase intracellular oxidative damage by promoting the production of intracellular ROS in bacteria, eventually causing their death.

The bacterial outer membrane is fluid, which is critical for their proliferation and survival (26). We found that the combination treatment resulted in a sharp decrease in cell membrane fluidity, and this process disrupted bacterial intracellular homeostasis, including PMF dissipation ($\Delta$pH) and membrane depolarization ($\Delta\Psi$) (Fig. 7C to E; Fig. S4C to E and S6A to C). In addition, the antimicrobial activity of baicalein decreased as the pH (5.5 to 9.0) of Mueller-Hinton broth (MHB) increased, because intracellular $\Delta$pH responds to changes in extracellular pH (Fig. S6D). The uptake of aminoglycoside antibiotics depends on $\Delta\Psi$ (27). The MIC of gentamicin decreased with an increase in the concentration of baicalein (Fig. S6E). The depolarization of bacterial cell membranes further decreased the production of intracellular ATP (Fig. 7F; Fig. S4F and S6F). As membrane efflux pumps require energy from ATP, in the combination treatment, ethidium bromide (EtBr) efflux and biofilm formation were lower (Fig. 7G and H; Fig. S4G and H and S6G). These findings indicated that baicalein combined with doxycycline can inhibit bacterial multidrug efflux and biofilm formation.

**Baicalein mitigates LPS-induced inflammatory response *in vitro*.** Nonobligately intracellular bacteria, such as *A. baumannii* and *K. pneumoniae*, might avoid host immune defense clearance and antibiotic killing by invading and surviving in host cells (28, 29). Baicalein (125 $\mu$g/mL) in combination with doxycycline (0 to 512 $\mu$g/mL) decreased the intracellular bacterial load in 16HBE cells more effectively than doxycycline alone (Fig. 8A

**FIG 2** Legend (Continued)
checkerboard assay. Darker blue areas indicate higher cell density and lower inhibition caused by combinatorial therapy. (B) Time-dependent killing curve of *A. baumannii* AB145 and *K. pneumoniae* KP1 with a combinatorial treatment of baicalein and doxycycline. Pathogens were grown to the exponential phase in MHB and then were treated with PBS, a sub-MIC of doxycycline (DOX, 64 $\mu$g/mL), a sub-MIC of baicalein (125 $\mu$g/mL), or both (DOX+baicalein; 64 $\mu$g/mL + 125 $\mu$g/mL) for 24 h. (C) Viable counts of bacteria after treatment with baicalein (62.5 $\mu$g/mL) alone or in combination with doxycycline (1/4× MIC or 1/2× MIC) for 16 h. The data are mean ± SD, and differences were assessed using nonparametric one-way ANOVA; ***, $P < 0.001$. (D) The LIVE/DEAD BacLight bacterial viability kit was used to evaluate the percentage of live and dead bacteria. (E) Representative fluorescent images of *A. baumannii* AB145 and *K. pneumoniae* KP1 live (green fluorescence) and dead (red fluorescence) bacterial cells after treatment with baicalein (62.5 $\mu$g/mL) and doxycycline (32 $\mu$g/mL) alone or in combination for 8 h. Bar, 50 $\mu$m. (F) Adding baicalein (62.5 $\mu$g/mL, 1/4× MIC) prevented doxycycline resistance in *A. baumannii* AB145 and *K. pneumoniae* KP1 *in vitro*. Resistance was acquired during serial passage in 0.25× MIC of doxycycline. The data are based on three biological replicates and presented as the mean ± SD.

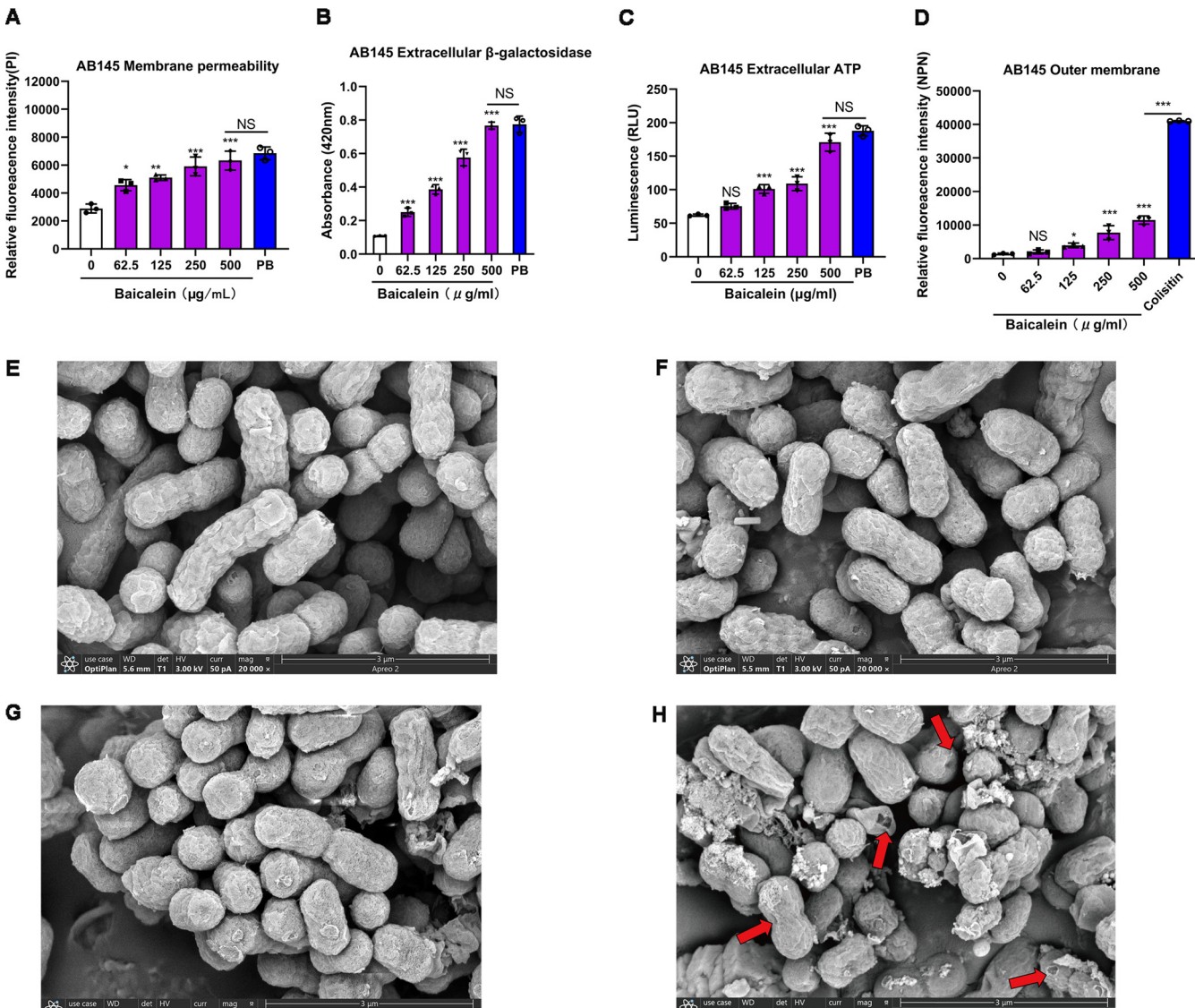

**FIG 3** Baicalein exerts antibacterial activity through the inner and outer membranes of *A. baumannii*. (A) The permeability of the cytoplasmic membrane of *A. baumannii* AB145 increased after treatment with baicalein (0 to 500 $\mu$g/mL) or PB (16 $\mu$g/mL) for 30 min and probing with 10 nmol/L of PI. (B and C) The release of bacterial contents, including extracellular (B) $\beta$-galactosidase (molecular weight [MW], 130 kDa) and (C) ATP (MW, 507 Da), increased after treatment with baicalein (0 to 500 $\mu$g/mL) for 30 min. (D) The permeability of the outer membrane of *A. baumannii* AB145 increased after treatment with baicalein (0 to 500 $\mu$g/mL) or colistin (4 $\mu$g/mL) for 30 min and probing with 10 $\mu$mol/L of NPN. The data in panels A to D represent three biological replicates, and the error bars represent SD. The *P* values were determined by performing nonparametric one-way ANOVA. NS, not significant; *, $P < 0.05$; **, $P < 0.01$; ***, $P < 0.001$. (E to H) Morphological changes in *A. baumannii* AB145 after 1-h treatment with PBS (E), a sub-MIC of doxycycline (64 $\mu$g/mL) (F), a sub-MIC of baicalein (125 $\mu$g/mL) (G), or the combination (H), visualized using SEM. Bar = 3 $\mu$m. Red arrows indicate the destroyed outer membrane.

and B). Then, we used LPS to evoke an inflammatory reaction that simulates a bacterial infection in Raw264.7 cells. The exogenous addition of baicalein strongly suppressed the generation of LPS-mediated proinflammatory factors *in vitro* (Fig. 8C to F). Thus, baicalein treatment can inhibit bacterial invasion of host cells and modulate the host immune system to enhance immunity against certain infections.

**Baicalein improves doxycycline efficacy *in vivo*.** Baicalein combined with doxycycline showed an effective antibacterial activity *in vitro* (Fig. 8G), which prompted further analysis of this combinatorial treatment in infection models. We used a neutropenic mouse lung infection model to evaluate the effects of this combinatorial treatment *in vivo*. Following the addition of baicalein, the bacterial loads in the bronchoalveolar lavage fluid and mouse organs were the lowest in the combination treatment group (Fig. 9). Additionally, the levels of four inflammatory cytokines (interleukin 1$\beta$ [IL-1$\beta$], gamma

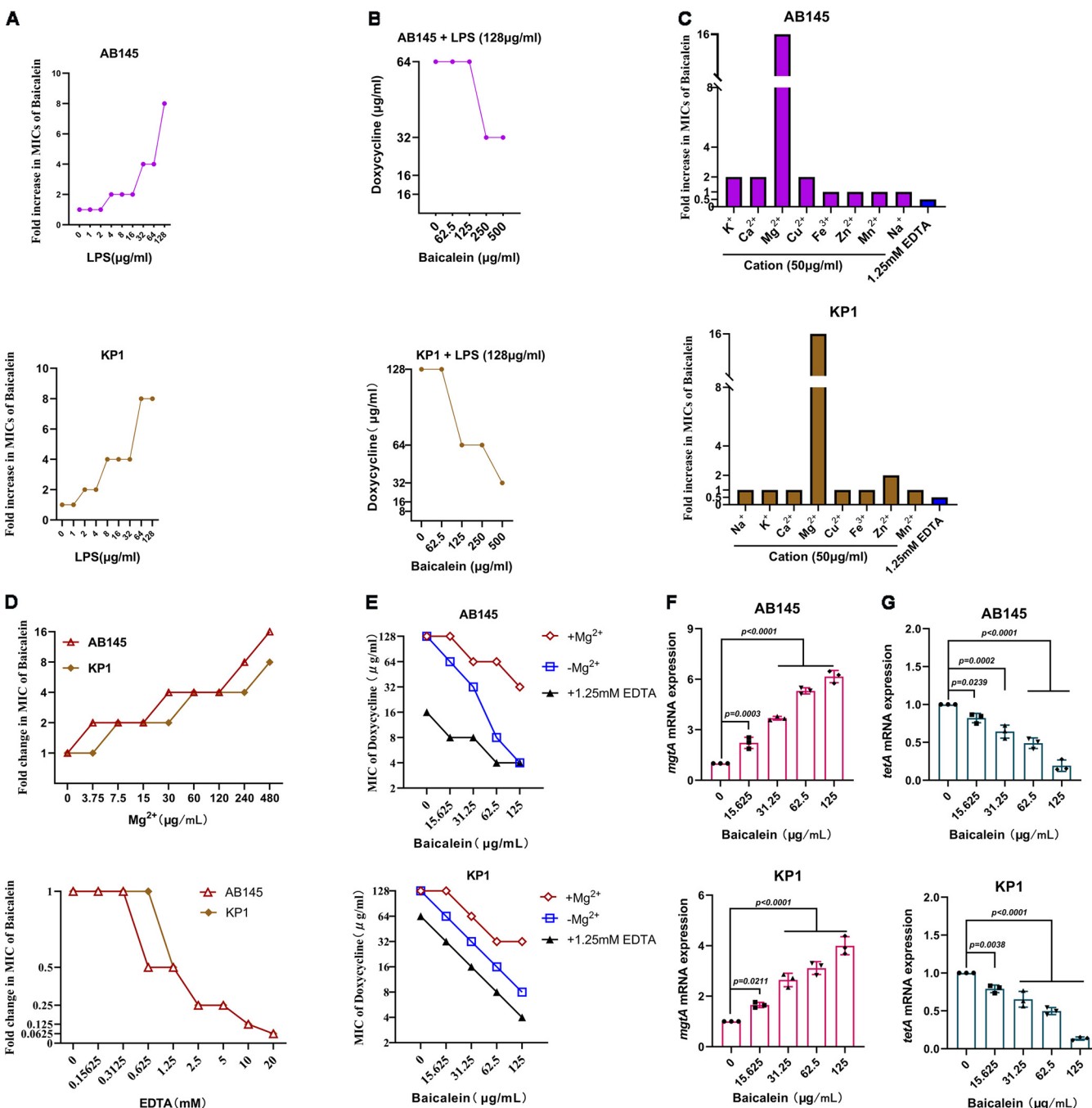

**FIG 4** Baicalein competes with $Mg^{2+}$ to bind to lipid A. (A) Exogenous addition of *A. baumannii*-derived or *K. pneumoniae*-derived LPS (0 to 128 $\mu$g/mL) impaired the antibacterial activity of baicalein against AB145 and KP1 in a dose-dependent manner, determined by a checkerboard microdilution assay. (B) Purified LPS (128 $\mu$g/mL) reduced the antibacterial activity of baicalein combined with doxycycline. (C) The effects of different cations (50 $\mu$g/mL) and EDTA (1.25 mM) on the antibacterial activity of baicalein against AB145 and KP1. (D) $Mg^{2+}$ inhibited the antibacterial activity of baicalein against AB145 and KP1 in a dose-dependent manner; EDTA was used as a positive control. (E) The exogenous addition of $Mg^{2+}$ (480 $\mu$g/mL) reduced the antibacterial activity of baicalein combined with doxycycline. (F) The expression of the *mgtA* mRNA, which belongs to the PhoPQ two-component system, presents increased with increasing baicalein does. (G) As determined by RT-PCR analysis, baicalein inhibited the transcription of *tetA* in a dose-dependent manner. The data are means $\pm$ SD, and the differences were assessed using nonparametric one-way ANOVA.

interferon [IFN-$\gamma$], tumor necrosis factor alpha [TNF-$\alpha$], and IL-8) in serum were considerably lower after 48 h of pulmonary infection in the combination treatment group than in other groups, which was confirmed by the alleviated pulmonary pathology (Fig. S7). These results indicated that baicalein combined with doxycycline can inhibit bacterial invasion into host cells and reduce the host inflammatory response to these infections *in vivo*.

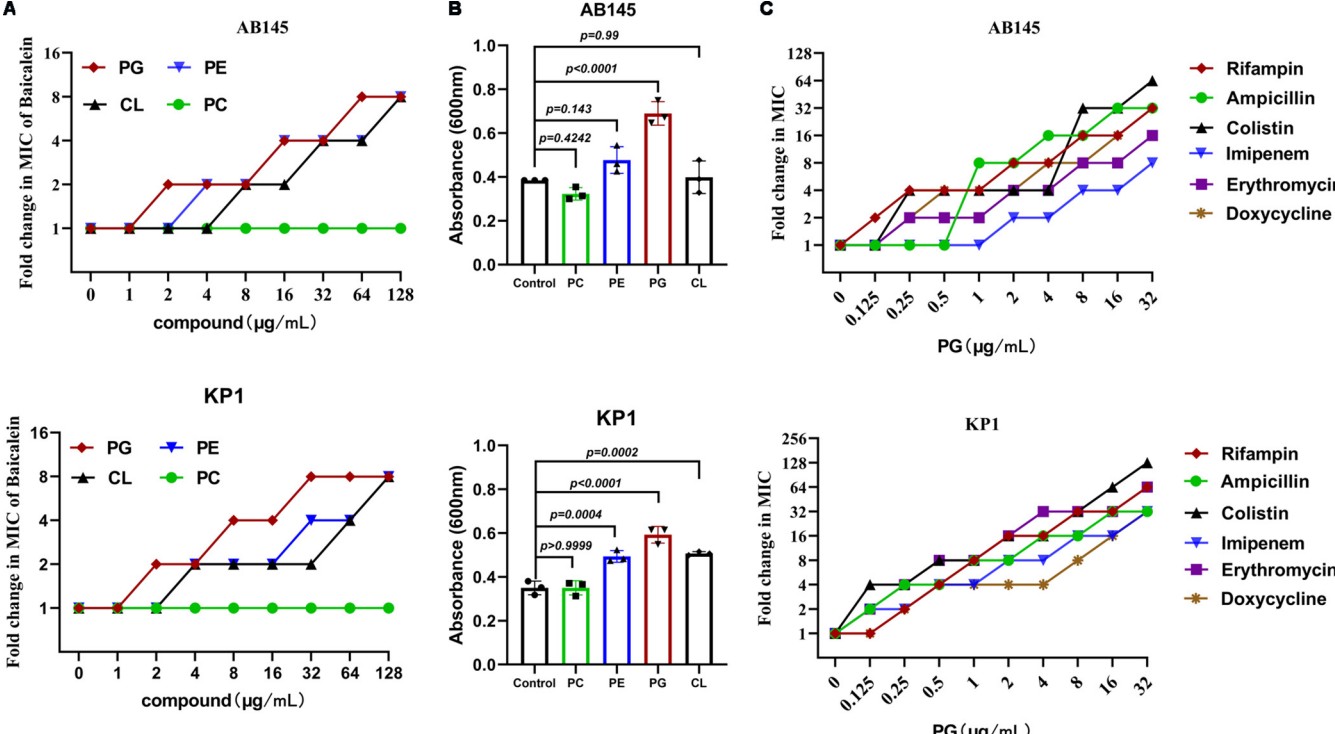

**FIG 5** Phospholipids reduce the antibacterial activity of baicalein against *A. baumannii* and *K. pneumoniae*. (A) The MICs of baicalein against AB145 and KP1 increased in the presence of phospholipids (0 to 128 $\mu$g/mL), as determined by a checkerboard microdilution assay. PC was used as a control. (B) PG decreased the antibacterial activity of baicalein against AB145 and KP1. Turbidity of the mixtures of baicalein (sub-MIC, 50 $\mu$L) with various phospholipids (5 mg/mL, 50 $\mu$L) at 600 nm. (C) PG blocked the antibacterial activity of baicalein combined with multiple antibiotics (rifampin, ampicillin, colistin, imipenem, erythromycin, and doxycycline) against AB145 and KP1. A sub-MIC of baicalein was added to the bacterial suspension in the presence of PG (0 to 32 $\mu$g/mL) combined with each antibiotic. The data are means ± SD, and the differences were determined by nonparametric one-way ANOVA.

## DISCUSSION

Baicalein is a bioactive flavonoid extracted from the roots of *S. baicalensis* Georgi. Previous studies have reported that baicalein not only exhibits synergistic effects with antibiotics (15–17) but also has anti-inflammatory effects (30). However, the detailed mechanism of baicalein combined with antibiotics has a therapeutic effect on bacterial infections, which is not fully understood. In this study, we focused on the bacterial cell membrane, metabolism, efflux pumps, and biofilm formation in two *tetA*-positive MDR Gram-negative clinical isolates, including *A. baumannii* AB145 and *K. pneumoniae* KP1, to determine the mechanism by which baicalin resensitizes MDR Gram-negative bacteria to doxycycline. Baicalein bound to phospholipids and inhibited $Mg^{2+}$ from attaching to lipid A to destroy the cell membrane of Gram-negative bacteria. Baicalein also altered bacterial metabolism by enhancing the TCA cycle activity, triggering ROS production, and inhibiting multidrug efflux pumps and biofilm formation. Additionally, baicalein prevented the entry of bacteria into the host cell and attenuated host inflammatory responses to bacterial invasion.

Baicalein exhibited synergistic antibacterial activity by disrupting the inner and outer membranes of Gram-negative bacteria. We found that baicalein inhibited $Mg^{2+}$ bound to lipid A to destroy the outer membrane and targeted phospholipids in the cytoplasmic membrane of the Gram-negative bacteria. Lipid A is a lipid LPS component that can electrostatically bind to divalent cations and provide stability to the outer membrane, which prevents the entry of numerous efficient antibacterial drugs (31). Several studies have suggested that phospholipids are essential targets for antibiotic therapy in Gram-negative bacteria (32, 33). However, the cell membranes of Gram-negative bacteria contain only a small amount of cardiolipin, while phosphatidylethanolamine is abundant in mammalian cells. This limits their utility as therapeutic targets for bacterial infections. Phosphatidylglycerol is the main component of the phospholipid

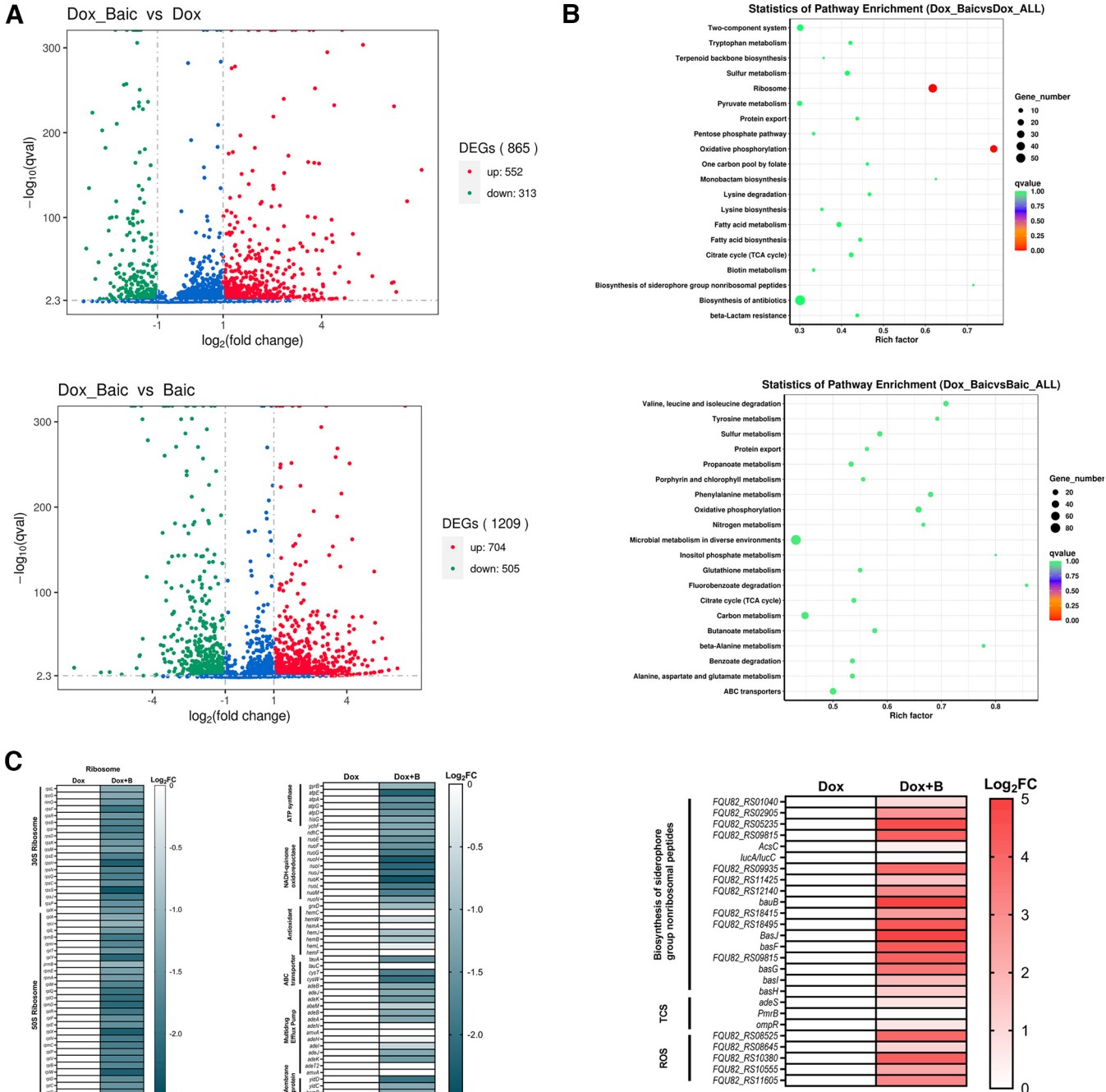

**FIG 6** Transcriptomic analysis of *A. baumannii* AB145 after exposure to different treatments. Volcano plots illustrate the RNA sequencing transcriptome (A) and KEGG enrichment analysis (B) of the DEGs in AB145 after treatment with doxycycline (64 $\mu$g/mL), baicalein (125 $\mu$g/mL), or their combination for 4 h. In panel A, the *x* and *y* axes indicate the changes in the expression and the corresponding degree of statistical significance. (C) The selected DEGs were associated with ribosomes, oxidative phosphorylation, antioxidants, multidrug efflux pumps, ABC transporters, membrane proteins, the biosynthesis of siderophore group nonribosomal peptides, ROS, and the two-component system (TCS). DOX, doxycycline alone; DOX+B, combination of doxycycline and baicalein. The data are from three biological replicates.

layer of the plasma membrane, but its content in mammalian cell membranes is low (34). Thus, phosphatidylglycerol might be an ideal antibacterial target. The effectiveness and low toxicity of baicalein confirmed this hypothesis.

Baicalein altered bacterial metabolism and decreased multidrug efflux pump activity to potentiate the antibacterial activity of doxycycline. The activity of the TCA cycle and ROS production increased in the presence of baicalein in *A. baumannii* and *K. pneumoniae*. The production of ROS is a common mechanism by which bactericidal

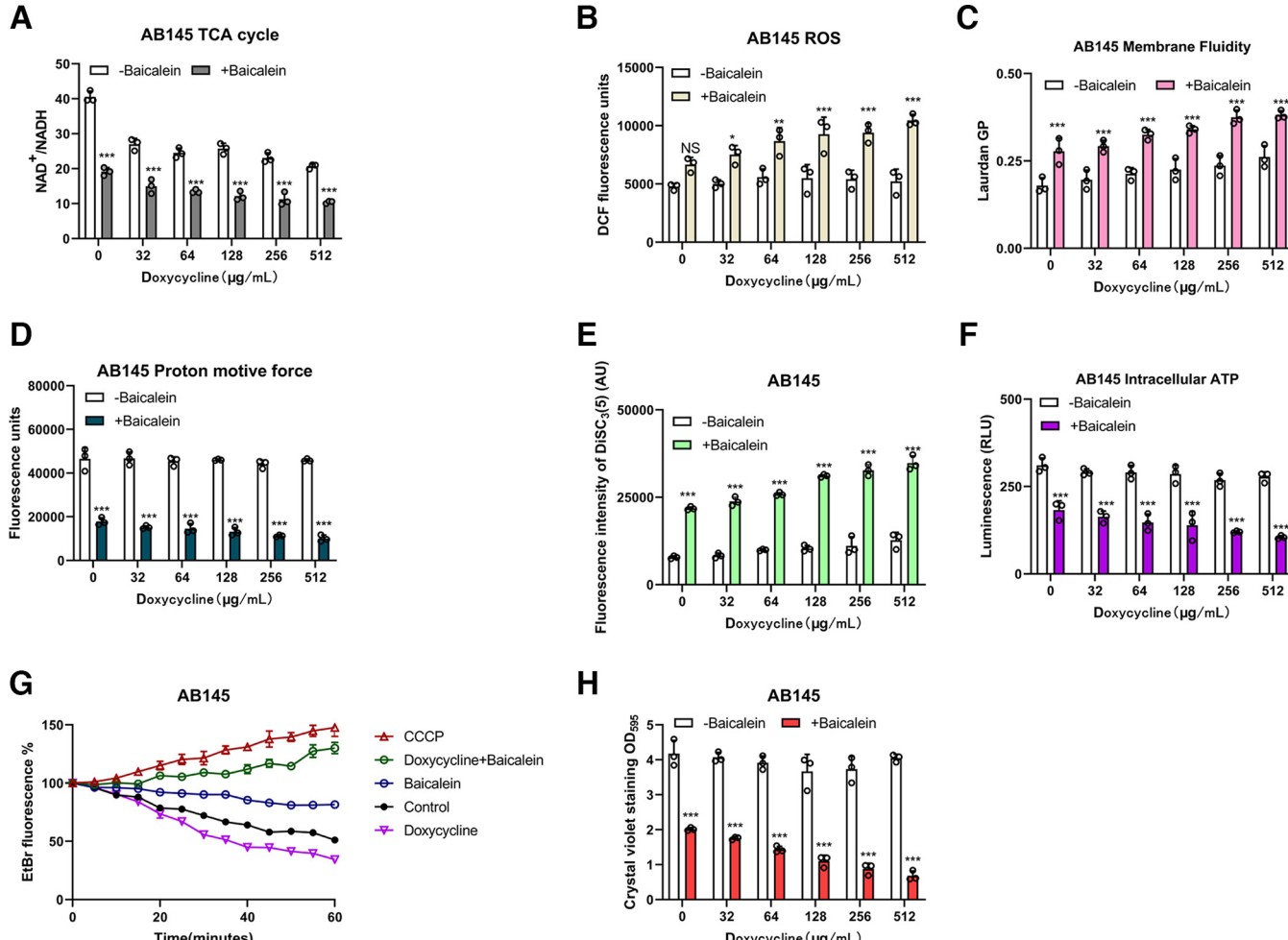

**FIG 7** Synergistic mechanisms of baicalein combined with doxycycline against *A. baumannii*. (A) Combining baicalein (125 µg/mL) and doxycycline enhanced the activity of the TCA cycle. (B) Adding baicalein (125 µg/mL) to doxycycline increased ROS generation. (C) Membrane fluidity of AB145 decreased after combined treatment. (D) Baicalein (125 µg/mL) combined with doxycycline promoted PMF dissipation, assessed by monitoring the fluorescence intensity of BCECF-AM-probed AB145 cells. (E) The combined treatment promoted bacterial $\Delta\Psi$ depolarization, evaluated by measuring the fluorescence intensity of $DiSC_3(5)$. (F) Decrease in the intracellular ATP levels in AB145 after the colocalization of baicalein with doxycycline. (G) Inhibition of EtBr efflux pumps by treatment with baicalein (125 µg/mL) combined with doxycycline (64 µg/mL) in AB145. The known efflux pump inhibitor CCCP ($10 \times 10^{-5}$ M) was used as a positive control. Untreated bacteria acted as the control. (H) Comedication with baicalein (125 µg/mL) suppressed biofilm formation in AB145. Three biologically independent experiments were conducted, and the data are means ± SD. Nonparametric one-way ANOVA was performed to determine the $P$ values (NS, not significant; *, $P < 0.05$; **, $P < 0.01$; ***, $P < 0.001$).

antibiotics induce cell death (35). Meanwhile, bacteria have evolved efflux pump systems to defend themselves against antibiotics. After baicalein treatment, the PMF and ATP levels, which provide the energy required for the drug efflux pump, were reduced. Additionally, disturbances in the intracellular environment caused by damage to the bacterial cell membrane can lead to further PMF depletion (36). The inactivation of efflux pumps can disrupt biofilm formation and pathogenicity in Gram-negative bacteria (37, 38). Baicalein alone or combined with vancomycin has an anti-quorum sensing effect (30, 39). Quorum sensing plays an essential role in biofilm formation, consistent with our results that biofilm formation decreases in the presence of baicalein. Our findings suggested that the above-mentioned mechanisms enhanced doxycycline accumulation in bacterial cells and caused the death of bacteria.

Baicalein combined with doxycycline significantly increased the elimination of *tetA*-positive Gram-negative bacteria *in vivo*. This study showed that the combined treatment mitigated LPS-elicited endotoxin-mediated inflammatory responses. A study showed that baicalein can suppress ROS generation and JAK/STAT activation in macrophages, thus decreasing LPS-mediated inflammation (40). Baicalein can also decrease the activation of the NF-κB and

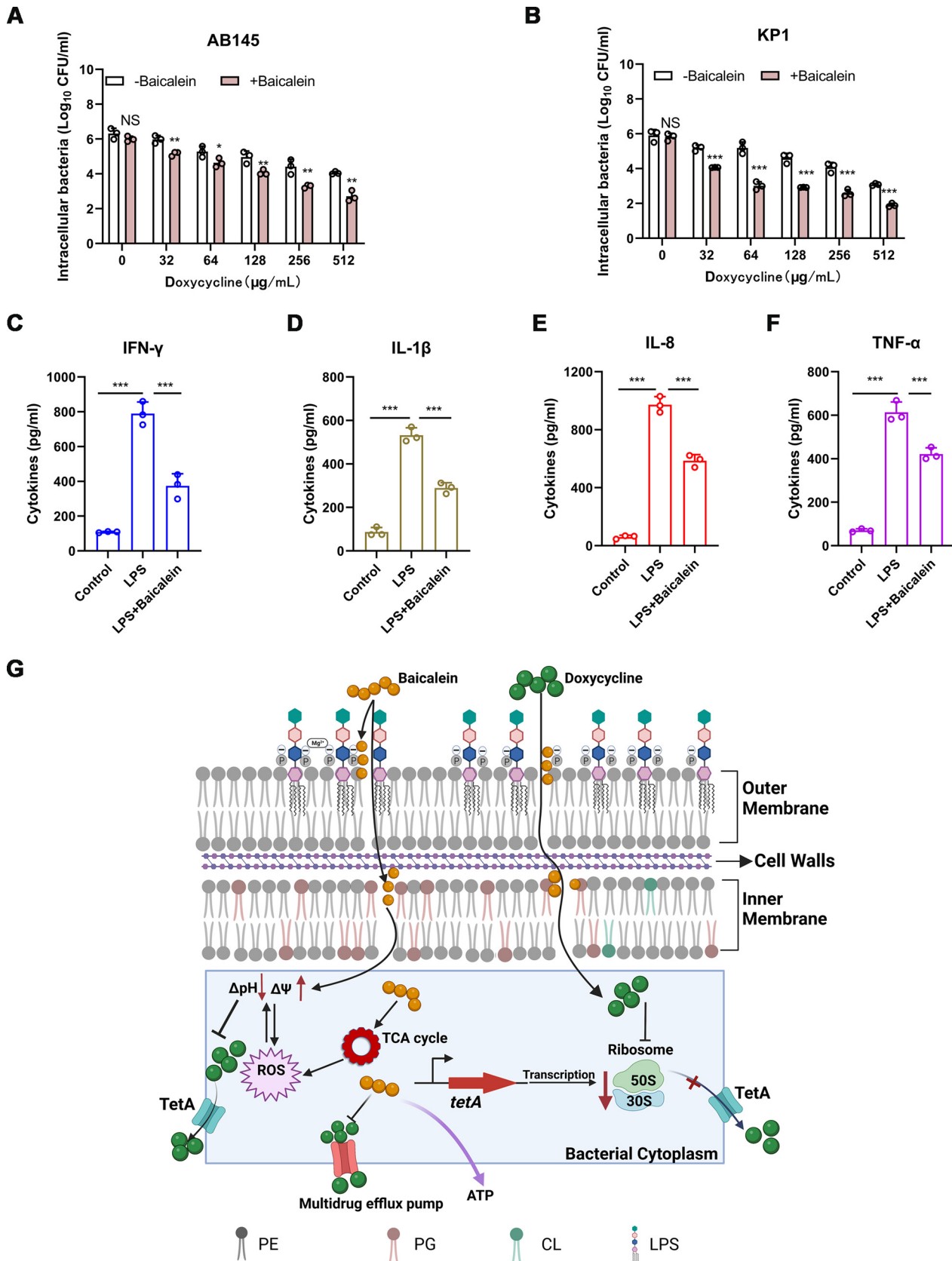

**FIG 8** Baicalein attenuated the inflammatory response induced by LPS. (A and B) The combinatorial treatment of baicalein (125 μg/mL) and doxycycline (0 to 512 μg/mL) for 3 h reduced the intracellular bacterial load of *A. baumannii* AB145 or *K. pneumoniae* KP1 in 16HBE cells

MAPK pathways to mitigate the inflammatory response in macrophages infected with *P. aeruginosa* (30). However, further studies need to be performed to elucidate the immunomodulatory mechanism by which baicalein resists bacterial infections.

In conclusion, our findings are the first to show that baicalein sensitizes *tetA*-positive MDR Gram-negative pathogens to doxycycline by direct engagement with phospholipids and lipopolysaccharides, thus disturbing the permeability of the inner and outer cell membranes of Gram-negative bacteria. Baicalein addition perturbed the intrinsic antibiotic tolerance mechanisms of Gram-negative bacteria and reduced the host inflammation response triggered by these bacterial infections. Nevertheless, some limitations of this study should be addressed in the future. We found that baicalein can significantly reduce the MIC of doxycycline for Gram-negative isolates; however, an excessive concentration of baicalein was used in the combination, which makes it challenging to reach effective drug concentrations in the plasma of patients. Future research should focus on optimizing the pharmacologic properties of baicalein to enhance its antimicrobial activity and bioavailability as an independent antibiotic or potentiator.

## MATERIALS AND METHODS

**Bacteria, cells, and chemicals.** All strains used in this study are presented in Table 2. Based on previous studies, *tetA* and *tetB* were determined by multiplex PCR in clinical strains (41, 42). *Acinetobacter baumannii* AB43Δ*crispr-cas* (XDR) and the quality control strain *A. baumannii* ATCC 19606 were maintained in our laboratory (20). All strains were cultivated in MHB (Oxoid Ltd., Cambridge, UK) at 37°C and 220 rpm unless stated otherwise. The 16HBE and THP-1 cells were maintained in RPMI 1640 medium (Gibco) supplemented with 10% heat-inactivated fetal bovine serum (FBS) (Invitrogen) and 50 $\mu$M $\beta$-mercaptoethanol. RAW264.7 and A549 cells were cultured in Dulbecco's modified Eagle medium (DMEM) (Gibco) with 10% heat-inactivated FBS (Invitrogen). Baicalein and antibiotics were obtained from Selleckchem (Shanghai, China). According to methods published by the Clinical and Laboratory Standards Institute (CLSI, USA), baicalein was dissolved in dimethyl sulfoxide (DMSO) with a final concentration of less than 1%.

**Antibacterial susceptibility and synergy testing.** Serial 2-fold dilutions of each antimicrobial agent were performed to determine microdilution MICs in the broth, following the CLSI 2021 guidelines (43). Tested antibiotics were added to triplicate wells of 96-well flat-bottomed tissue culture plates (Corning), and 2-fold serial dilution was performed, followed by the addition of the prepared bacterial inoculum (0.5 × 10⁶ CFU/mL). After incubation for 18 h at 37°C, the minimum concentration of antibiotic without visible bacterial growth was recorded as the MIC. HCl or NaOH was added to the medium to adjust the pH to 5 to 9.

Checkerboard assays were performed on a 6-by-10 dose-response combination matrix to determine the synergy *in vitro*. FICI was calculated as the MIC of the combination divided by the MIC of each compound separately (44). The calculated FICIs were defined as synergistic (FICI ≤ 0.5), additive (0.5 < FICI < 1), indifferent (1 ≤ FICI < 4.0), or antagonistic (FICI ≥ 4.0).

**Bacterial growth inhibition assay.** An overnight culture was diluted with sterile MHB at 1:100 and incubated for 4 h (exponential phase) at 37°C with shaking at 220 rpm. Then, sub-MICs of baicalein (125 $\mu$g/mL) and doxycycline (64 $\mu$g/mL) were added to the bacterial suspension separately or in combination and incubated for 24 h. At the corresponding time, differently treated cocultured aliquots of the bacterial culture were collected, serially 10-fold diluted, and then spotted onto Mueller-Hinton agar medium (MHA; Oxoid Ltd., Cambridge, UK) plates for calculating the number of CFU. The viability of the bacteria was determined by the number of CFU per milliliter after incubation for 16 h (45).

**Bacterial live/dead staining.** Bacteria cultured in the logarithmic phase of growth were diluted to 1 × 10⁶ CFU/mL in MHB. The bacterial suspensions were incubated with baicalein (62.5 $\mu$g/mL) or doxycycline (32 $\mu$g/mL) alone or in combination for 8 h. The cells were then harvested by centrifugation and resuspended in sterile phosphate-buffered saline (PBS). LIVE/DEAD BacLight bacterial viability kits (Molecular Probes) containing SYTO9 and PI were used to distinguish between live and dead bacteria. The samples were visualized using a Zeiss Axio Scope A1 upright microscope, and the images were processed for merging using the ImageJ software (46).

**FIG 8** Legend (Continued)

compared to doxycycline treatment alone. (C to F) Acute inflammation triggered by bacterial LPS (1 $\mu$g/mL) was alleviated by baicalein (125 $\mu$g/mL). After pretreatment with baicalein (125 $\mu$g/mL) for 30 min, Raw264.7 cells were stimulated with LPS for 24 h. ELISA was performed to estimate the cytokine levels in culture samples after incubation. Three biologically independent experiments were conducted, and the data are means ± SD. The $P$ values were evaluated by conducting unpaired $t$ tests between groups or one-way ANOVA among multiple groups (NS, not significant; *, $P < 0.05$; **, $P < 0.01$; ***, $P < 0.001$). (G) Mechanism of action of baicalein combined with doxycycline against *tetA*-positive Gram-negative bacteria. Baicalein can restore the susceptibility to doxycycline of MDR Gram-negative bacteria via membrane-mediated processes by interacting with LPS in the outer membrane and targeting phospholipids in the cytoplasmic membrane. These processes cause membrane depolarization and malfunction, which in turn leads to metabolic alterations such as an increase in the TCA cycle activity and oxidative damage. Doxycycline can also bind to bacterial ribosomes and inhibit the production of bacterial proteins such as TetA.

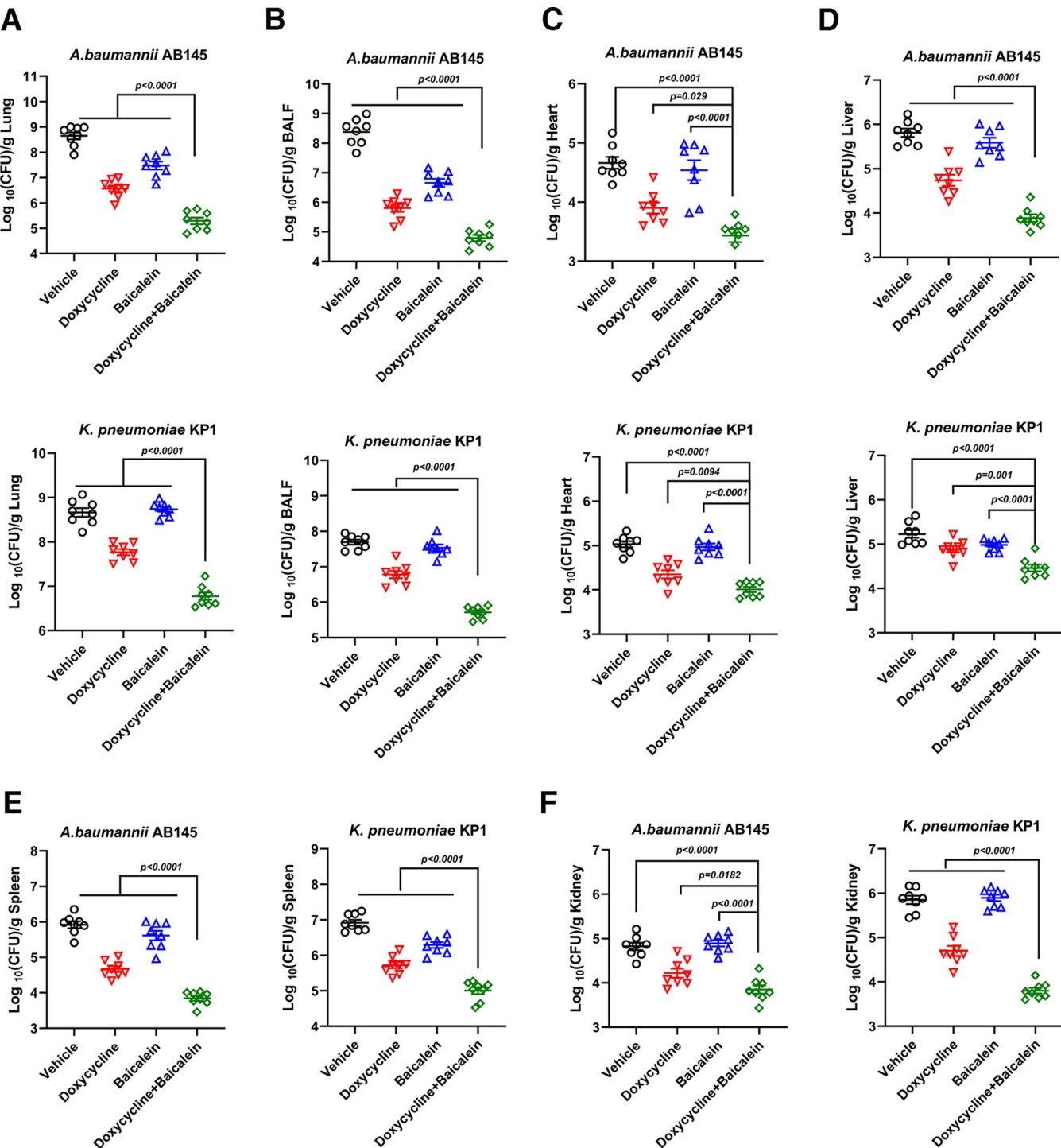

**FIG 9** Combinatorial therapy decreased bacterial load in the neutropenic mouse lung infection model. Neutropenic female BALB/c mice (8 per group) were administered a nonlethal dose of Gram-negative bacteria ($1.0 \times 10^6$ CFU) to the lungs, and 2 h later, the mice were treated intraperitoneally with a single dose of baicalein (50 mg/kg), doxycycline (50 mg/kg), or the combination or with PBS; all mice were euthanized by cervical dislocation after 48 h. In the lung (A), bronchoalveolar lavage fluid (BALF) (B), heart (C), liver (D), spleen (E), and kidney (F), bacterial burdens ($\log_{10}$ CFU of AB145 or KP1) were detected. Three biologically independent experiments were conducted, and data are means ± SD. The $P$ values were determined by performing two-sided Mann-Whitney U tests.

**Safety evaluation.** Defibrinated sheep blood cells (8%) were coincubated with different concentrations of combinations for 1 h. After incubation, the optical density at 576 nm ($OD_{576}$) was used to determine hemolytic activity, and the hemolytic rate (as a percentage) was calculated as follows: $[(OD_{sample} - OD_{blank})/(OD_{0.2\%TritonX-100} - OD_{blank})] \times 100$ (47). The cytotoxic effects on A549, RAW264.7, and THP-1 cells were assessed by performing the Cell Counting Kit-8 (CCK-8) assay following specific protocols (Beyotime,

Shanghai, China). When the cells reached 80% to 90% confluence, the medium was discarded, and the cells were washed with PBS three times and cultured in the corresponding serum-free medium (48–50). Doxycycline (0 to 128 $\mu$g/mL) in combination with baicalein (2 to 250 $\mu$g/mL) was added to 96-well plates along with $1 \times 10^4$ cells, which were cultured for 24 h at 37°C. After 24 h of culture, the CCK-8 solution (10 $\mu$L/well) was added to each well. The OD values were determined at 450 nm to evaluate the cytotoxic effect of doxycycline combined with baicalein.

**Resistance development studies.** Bacterial cultures ($0.5 \times 10^6$ CFU/mL) containing doxycycline ($0.25\times$ MIC) alone or in combination with baicalein (62.5 $\mu$g/mL) were incubated at 220 rpm for 24 h (37°C). Subsequently, the strains with visible growth ($OD_{600} \geq 0.3$) were passaged on antibiotic-free MHA plates for 24 h, and the corresponding MICs of doxycycline were detected by the 2-fold serial dilution method described above. The bacterial suspension ($OD_{600} = 0.5$) was diluted 1:100 with MHB and incubated in the presence of doxycycline (1/4 MIC) alone or in combination with baicalein (62.5 $\mu$g/mL) for the next generation. This step was performed repeatedly for 30 days (51). Sangon Biotech sequenced all mutant strains. All sequence alignment and single-nucleotide polymorphism (SNP) identification were carried out with MEGA 5.0 software (52).

**Cell inner and outer membrane integrity assay.** Fresh bacterial cultures ($OD_{600} = 0.5$) were incubated with the indicated concentrations of baicalein or polymyxin B (PB; 16 $\mu$g/mL) at room temperature for 30 min. Next, the bacterial suspensions were mixed with 10 nmol/L of PI (Solarbio, Beijing, China) to detect the permeabilization of the inner membrane, and then the suspensions were coincubated for 30 min in the dark. The fluorescence was detected at emission/excitation wavelengths of 615/535 nm (46). In the fluorescence or luminescence assay, the corresponding concentrations of drugs (without cells) were incubated with probes as blank controls, and the relative fluorescence intensity was calculated as the value of the analyte minus the value of the blank control.

For the outer membrane permeability assay, 10 $\mu$mol/L of NPN (Solarbio, Beijing, China) was used and the assay was performed following the same procedure, with colistin (4 $\mu$g/mL) used as a positive control. The fluorescence intensity was measured at emission/excitation wavelengths of 420/350 nm (44).

**Extracellular $\beta$-galactosidase measurement.** Different concentrations of baicalein or PB (16 $\mu$g/mL) were coincubated with bacterial suspension ($OD_{600} = 0.5$) for 30 min (37°C). Then, 190 $\mu$L of supernatants (12,000 $\times$ $g$, 15 min, 4°C) was incubated with 2-nitrophenyl-$\beta$-D-galactopyranoside (ONPG; 3 mmol/L; Sigma) for 30 min, and $OD_{420}$ was detected (53).

**Measurement of ATP content.** The bacterial suspension ($OD_{600} = 0.5$) was coincubated with the indicated concentrations of drugs for 30 min. Then, the lysed bacterial precipitates and bacterial supernatants (12,000 $\times$ $g$, 15 min, 4°C) were used to determine the intracellular and extracellular ATP content, respectively (Beyotime; catalog no. S0027).

**SEM.** Doxycycline (64 $\mu$g/mL), baicalein (125 $\mu$g/mL), or their combination was used to treat bacterial suspension ($OD_{600} = 0.5$) for 1 h, followed by overnight incubation with 2.5% glutaraldehyde at 4°C. After thorough rinsing three times using PBS, the bacteria were subjected to gradual dehydration in the sequence of 30%, 50%, 70%, 80%, 90%, and 95% ethanol for 10 min each and 100% ethanol for 10 min three times. Subsequently, the samples were dried in a critical point dryer, coated with gold-palladium using an ion sprayer (54), and then observed with SEM (GeminiSEM 300).

**LPS, cationic-ion, and phospholipid assays.** Bacterial LPS was extracted using a kit from iNtRON Biotech (Gyeonggi-Do, South Korea). Next, LPS, baicalein, doxycycline, and bacterial suspensions ($5 \times 10^8$ CFU/mL) were added to a 96-well plate, followed by 18 h of incubation at 37°C. The checkerboard microdilution assay was performed to assess how LPS (0 to 128 $\mu$g/mL) affected the antibacterial activity of baicalein. MIC assays were performed to determine the impact of different cationic ions (50 $\mu$g/mL), including KCl, $CaCl_2$, $MgCl_2$, $CuSO_4$, $FeCl_3$, NaCl, $MnSO_4$, and $ZnSO_4$ (Aladdin, Shanghai, China), on the antibacterial activity of baicalein on *A. baumannii* or *K. pneumoniae*. PG (Sigma-Aldrich; 841188P; $\geq$99%), PC (Sigma-Aldrich; catalog no. 840051P; $\geq$99%), CL (Sigma-Aldrich; 841199P; $\geq$99%), and PE (Sigma-Aldrich; 840027P; $\geq$99%) were dissolved in methanol. The checkerboard microdilution assay was performed to determine the effects of different phospholipid concentrations (0 to 128 $\mu$g/mL) on the antimicrobial activity of baicalein in MHB.

**RT-PCR and transcriptomic analysis.** A bacterial suspension with an $OD_{600}$ of 0.5 was treated with the indicated concentrations of baicalein at 37°C for 30 min. Total RNA extraction and cDNA reverse transcription were conducted following a method described in another study (20). The mRNA expression of *tetA* (F, 5′-CTTGCCCCTAACCAACCGAACC-3′; R, 5′-AGGCCGTTTGCTTTCAGGGATC-3′) or *mgtA* (F, 5′-ACGTGAAGTGGCAATTGAGG-3′; R, 5′-TACCGTTACACCATGGGCAT-3′) in each sample was detected by 7500 Fast real-time PCR with SYBR green (Applied Biosystems, CA, USA) (44). The 16S rRNA served as an internal reference, and the $2^{-\Delta\Delta CT}$ method was applied to determine the relative gene levels.

*Acinetobacter baumannii* AB145 was grown in MHB to the early exponential phase. Next, bacterial cells were treated with 64 $\mu$g/mL doxycycline and 125 $\mu$g/mL baicalein independently or in combination for 4 h (55). Total RNA was extracted from these samples, and sequencing was performed using an Illumina HiSeq system (Motif Zhigu Biotechnology, Nanjing, China). DESeq2 was used to normalize raw read counts for estimating gene levels and determining differentially expressed genes (DEGs) with the thresholds of a false discovery rate (FDR) of <0.05 or a fold change (FC) of $\geq$2 with a $P$ value of <0.05. The Cuffdiff software (http://cufflinks.cbcb.umd.edu/) was used to analyze the differences between treatments.

**TCA cycle measurement.** Doxycycline (0 to 512 $\mu$g/mL) with or without baicalein (125 $\mu$g/mL) was added to bacterial suspensions ($OD_{600} = 0.5$). The bacterial pellets were washed with PBS after 1 h of incubation, followed by resuspension in 200 $\mu$L prechilled extraction buffer. The lysates were centrifuged at 12,000 $\times$ $g$ for 10 min at 4°C, and the supernatants were analyzed using the $NAD^+$/NADH assay kit with water-soluble tetrazolium salt-8 (WST-8, Beyotime, China) (56).

**Total ROS, SOD, and H$_2$O$_2$ measurement.** The bacterial suspension (OD$_{600}$ = 0.5) was treated with 0.01 mM 2′,7′-dichlorodihydrofluorescein diacetate (DCFH-DA; Invitrogen) to measure ROS levels for 30 min at 37°C. The bacterial suspension (190 $\mu$L) mixed with 10 $\mu$L of the indicated concentrations of baicalein was added to 96-well plates and incubated for 30 min. The fluorescence intensity was determined at excitation/emission wavelengths of 488/525 nm (57).

The hydrogen peroxide (H$_2$O$_2$) and SOD activities after treatment with baicalein, doxycycline, or their combination was determined with a hydrogen peroxide assay kit (Beyotime, Shanghai, China) and a total superoxide dismutase assay kit with WST-8 (Beyotime, Shanghai, China).

**Membrane fluidity assay.** Laurdan (10 nmol/L) was added to 5 × 10$^8$ CFU/mL of bacterial suspension and incubated at 37°C for 10 min. Next, the bacterial culture was mixed with various concentrations of baicalein or baicalein combined with doxycycline, colistin (4 $\mu$g/mL), or benzyl alcohol (50 mmol/L) for 30 min. The generalized polarization (GP) of laurdan was calculated as ($I_{440} - I_{490}$)/($I_{440} + I_{490}$), where $I_{440}$ and $I_{490}$ represent the emission intensity at 440 and 490 nm, respectively, when excitation was at 350 nm (58).

**PMF and membrane depolarization assay.** Bacterial suspension with an OD$_{600}$ of 0.5 was coincubated with 0.2 × 10$^{-6}$ M of 2′,7′-Bis (2-carboxyethyl-5(6)-carboxyfluorescein) (BCECF)-AM (acetoxymethyl ester) or 0.5 × 10$^{-3}$ mM of 3,3-dipropylthiadicarbocyanine iodide [DiSC$_3$(5); Sigma-Aldrich] for 30 min at 37°C. Then, 10 $\mu$L of different drugs, glucose (2.5 × 10$^{-5}$ M), or melittin (16 $\mu$g/mL) was added as a supplement and incubated for 30 min, and the fluorescence values of PMF and dissipated bacterial $\Delta\Psi$ were determined by excitation/emission at 500/522 nm and 622/670 nm, respectively (46).

**EtBr efflux and biofilm formation analysis.** The bacterial suspension (OD$_{600}$ = 0.5), EtBr (5 × 10$^{-6}$ M), drugs at specific concentrations, or the efflux pump inhibitor carbonyl cyanide $m$-chlorophenylhydrazone (CCCP; 10 × 10$^{-5}$ M) was incubated at room temperature. Then, efflux pump activity was monitored by taking readings at intervals of 5 min for 1 h at emission/excitation wavelengths of 600/530 nm.

Bacterial suspensions (5 × 10$^5$ CFU/mL) containing equivalent sub-MICs of different drugs were cultivated in triplicate on sterile flat-bottomed 96-well plates and incubated at 37°C for 24 h. The planktonic bacteria were removed, rinsed, and fixed with 4% paraformaldehyde for 20 min. Then, the wells were washed, air dried, and stained with 0.1% crystal violet for 20 min, followed by a repeated washing/drying process. Finally, 95% ethanol (200 $\mu$L) was added to the wells, and the OD$_{595}$ was measured.

**Intracellular bacterial content.** Bacteria (multiplicity of infection [MOI] = 100) were used to infect 16HBE cells, followed by coculturing using doxycycline (0 to 512 $\mu$g/mL) with or without a sub-MIC of baicalein at 37°C and 5% CO$_2$ for 3 h. Colistin (50 $\mu$g/mL) was then added to the cells for 15 min to remove extracellular bacteria. After rinsing twice with PBS, the cells were lysed using RPMI 1640 containing 0.1% Triton X-100, followed by inoculation of serial lysate dilutions in MHA to count CFU.

**Cytokine determination.** The levels of cytokines (TNF-$\alpha$, IL-8, IL-1$\beta$, and IFN-$\gamma$) were determined using commercial enzyme-linked immunosorbent assay (ELISA) kits (Beyotime, Shanghai, China) following specific protocols.

**Neutropenic mouse lung infection model.** Female BALB/c mice (6 to 8 weeks old; Comparative Medicine Centre of Yangzhou University, Jiangsu, China) were acclimatized for a week. Cyclophosphamide (150 and 100 mg/kg) was administered to 32 mice ($n$ = 8/group) on days 4 and 1 before infection for neutropenic induction. Next, the lungs of the mice were injected with 100 $\mu$L of bacterial suspension (1.0 × 10$^6$ CFU/mouse) (59). After 2 h of infection, the mice were administered an intraperitoneal injection of PBS, doxycycline (50 mg/kg), baicalein (50 mg/kg), or the combination (50 + 50 mg/kg). After 48 h, each mouse was euthanized by cervical dislocation. Peripheral blood serum samples of mice were collected and stored at −80°C for cytokine detection. The organs of the mice were dissected under sterile conditions, followed by homogenization, serial dilution, and inoculation in MHA for counting CFU. The left upper lung tissues of these mice were excised aseptically for hematoxylin-and-eosin (HE) staining. The mice were maintained according to the guidelines of the Administration of Affairs Concerning Experimental Animals issued by the State Council of the People's Republic of China (14 November 1988). The animal experiments were conducted following the instructions provided by the Guide for the Care and Use of Laboratory Animals (production permit number SCYK2022-0009; use permit number SYXK-2022-0044).

**Statistical analysis.** GraphPad Prism 8 was used for performing statistical analysis. The results were expressed as means and standard deviations (SD). Except when stated otherwise, unpaired $t$ tests and one-way analysis of variance (ANOVA) were performed to compare the data from two groups and more than two groups, respectively.

**Data availability.** Our transcriptome sequencing data were deposited in the NCBI database under the accession number PRJNA815471.

## SUPPLEMENTAL MATERIAL

Supplemental material is available online only.
**SUPPLEMENTAL FILE 1**, PDF file, 1.6 MB.

## ACKNOWLEDGMENTS

We thank Tao Zhu from the Affiliated Zhangjiagang Hospital of Soochow University, Yongqi Wang from the Lianyungang Oriental hospital, and Yawen Xu from Yangzhou CDC for providing the clinical strains.

We declare no competing interests.

Y.W. and G.L. were in charge of experimental design; Y.W., J.S., Z.Z., and J.Y. were responsible for experimental implementation; Y.W., J.S., Z.Z., J.Y., W.L., Y.Z., and P.Z. contributed to data analysis, figure/table plotting, and manuscript drafting; Y.W., T.G., and G.L. revised and approved the final version.

This work was funded by the National Natural Science Foundation of China (82073611, 82002186).

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
