## [Reviewer comments · Microbiology Spectrum]

Microbiology Spectrum

Baicalein Resensitizes Multidrug-Resistant Gram-negative Pathogens to Doxycycline

Yuhang Wang, Junfeng Su, Ziyang Zhou, Jie Yang, Wenjuan Liu, Yafen Zhang, Pengyu Zhang, Tingting Guo, and Guocai Li

Corresponding Author(s): Guocai Li, Yangzhou University Medical college

Review Timeline:

Submission Date:	November 17, 2022
Editorial Decision:	November 25, 2022
Revision Received:	January 24, 2023
Editorial Decision:	February 2, 2023
Revision Received:	March 31, 2023
Accepted:	March 31, 2023

Editor: Krisztina Papp-Wallace

Reviewer(s): The reviewers have opted to remain anonymous.

Transaction Report:

DOI: <https://doi.org/10.1128/spectrum.04702-22>

November 25, 2022

Prof. Guocai Li
Yangzhou University
Pathogen Biology and Immunology
11 Huai-hai Road
Yangzhou, Jiangsu 225001
China

Re: Spectrum04702-22 (Baicalein Resensitizes Multidrug-Resistant Gram-negative Pathogens to Doxycycline)

Dear Prof. Guocai Li:

Link Not Available

Sincerely,

Krisztina Papp-Wallace

Journals Department
Reviewer comments:

Reviewer #1 (Public repository details (Required)):

The transcriptomics data underlying Figure 6 are in NCBI, accession number PRJNA815471.

Reviewer #1 (Comments for the Author):

The manuscript by Want et al. examines the effect of baicalein on bacteria upon co-treatment with doxycycline. It is overall a thoughtful treatment of an important question. Baicalein is a flavonoid from a plant root that has been used in traditional Chinese medicine for treating bacterial infections. The authors show that baicalein potentiates multiple antibiotics in broth against Gram-

negative bacteria and then focus in on doxycycline. Through a series of assays, the authors show that co-treatment with dox and baicalein kills bacteria but does not kill or lyse mammalian cells. Baicalein damages *A. baumannii* outer and inner membranes and appears to specifically compete with magnesium to bind lipid A. The compound also likely interacts with bacterial lipids, especially PG. Transcriptomics comparing dox to dox + baicalein revealed many gene changes. Co-treatment also disrupted biofilms, reduced host responses to LPS, reduced mortality in wax-worms and reduced tissue colonization and inflammatory cytokines in mice. These data indicate that baicalein works through multiple mechanisms both at the level of the bacterium and on the host.

Major points

1) For the assays with fluorescent or luminescence probes, there need to be controls in which compound is incubated with the probe in the absence of cells at the concentrations of probe and compound used in the corresponding experiment. This is necessary to establish whether a compound on its own alters signal from the probe. These controls, or descriptions of the controls, were not found in the methods for Figure 7, for instance.

2) The transcriptomics are difficult to interpret without a baicalein-only sample.

Minor points

1) Figure 8 A and B - It would be more transparent to display Figure 8 A and B as bar graphs instead of line graphs, as the data are derived from different samples. Also, no error bars are visible, and the legend does not mention that there are error bars that are too small to be seen.

2) Sup Fig 1 E-H. Please put how long after treatment in the legend.

3) Ideally, bar graphs show individual data points so that readers can see data spreads.

Reviewer #2 (Comments for the Author):

This manuscript seeks to characterize the antibacterial activity of a natural product, baicalein, with particular focus on its ability to potentiate doxycycline activity in tetracycline-resistant Gram-negative bacteria. Baicalein is one of the active components in certain extracts used in Chinese traditional medicine that are thought to enhance the effectiveness of antibiotics in patients. As an isolated compound, it has been studied previously by others in order to evaluate its utility in various therapeutic indications, not all infectious disease. Its synergy with tetracyclines was studied in *S aureus* and was initially thought to result from inhibition of the TetK pump; it was then realized that baicalein could potentiate tetracycline in MRSA strains lacking TetK and that it could potentiate a variety of antibiotic classes in various bacterial species.

Li and colleagues undertook the current study with the idea that baicalein might restore the utility of doxycycline for treatment of Gram-negative infections. They found using checkerboard MICs that baicalein is technically synergistic with multiple antibiotics. The greatest effect is seen for doxycycline, where the MIC is reduced by up to 32 fold in some MDR strains of *A baumannii*. However, the resulting MIC is generally well above the clinical breakpoint. Moreover, these effects require fairly high concentrations of baicalein, 31.25 or 62.5 ug/ml. Although baicalein appears to be fairly well tolerated in mice, it seems unlikely that it will be feasible to reach these concentrations in plasma of patients treated with the proposed combination.

It is thus extremely unlikely that baicalein could be used clinically as an antibiotic partner to treat infections. Despite the very modest antibacterial activity of baicalein (MIC 250 ug/ml), one might feel it is worth studying, perhaps leading to novel approaches for designing agents with greater activity as standalone antibiotics or potentiators.

Much of the manuscript describes the attempt to determine the mechanism of action of baicalein and/or the mechanism by which it potentiates other antibacterial agents. Unfortunately, most of the mechanistic studies are not interpretable. In some studies, bacteria are treated with drugs for an hour or longer, with 4 hr for the gene expression studies. This is inappropriate, as the direct effects of antibacterial agents on bacterial physiology are generally apparent within 20-30 minutes. With longer times, the observed effects are indirect and shed no light on the molecular mechanism of the drug. Several experiments compared a subinhibitory treatment (single drug) to an inhibitory treatment (two drugs). It is likely that the effects seen are simply the effect of stress, growth inhibition or cell death. Any other inhibitory treatment might produce the same results.

I recommend shortening the manuscript substantially, focusing on the stronger parts of the work: in-vitro and in-vivo microbiology. I recommend delaying publication of the mechanistic studies until they have been conducted more carefully. These experiments should have included controls - i.e., comparator antibiotics with known mechanisms, and compounds previously characterized as OM disrupting agents, ionophores, or pore-forming agents. Bacterial viability should be monitored. Experimental conditions such as incubation times or concentration of reagents should be justified unless well supported by literature that is cited.

Specific comments:

1. Move tables S1 and S2 into the main text

2. Fig 1B. Please explain the spider-web plots. A table showing FICI for each antibiotic would be much easier to read and would allow the MIC of each antibiotic alone to be included. Why was this experiment done with strain AB43 Δ crispr-cas? What are the results for the drug-sensitive parent AB43?
3. Fig 1C and Table S2 show the same data for the same ten isolates of *A. baumannii*, but in a different order. Please present both in the same order, perhaps numerical by strain ID.
4. Fig 1D requires some explanation. What does "inhibition rate" mean? How was the experiment done?
5. Fig 2. Eliminate panel B. The untreated bacteria are growing and are clearly not in stationary phase
6. Fig 2 C-F, hemolysis and cytotoxicity. Low levels of both hemolysis and cytotoxicity are seen at very high concentrations of doxycycline (256 or 512 ug/ml), except in the presence of extremely high concentrations of baicalein (250 ug/ml). Lines 638-641 interpret these observations as showing baicalein protecting mammalian cells from detrimental effects of doxycycline. Is it generally thought that cytotoxicity of doxycycline is clinically relevant? If not, I suggest rewriting this section to focus on the lack of cytotoxicity and hemolysis for baicalein at concentrations that are likely to be achieved in vivo - both alone and in the presence of doxycycline.
7. Fig 2G and G, serial passage studies. At the start of the experiment, the MIC of doxycycline is 128 ug/ml for both AB145 and KP1. The plot shows the MICs increasing by 32 or 64 fold during passage in doxycycline unless baicalein was included. That would mean the final MICs are 4 or 8 milligrams/milliliter - is that correct? Suggest plotting actual MICs on the Y axis, rather than fold increase. Were these extremely high MICs still seen after the final culture had been passaged on drug-free agar? Did the resistant isolates have elevated MICs for other antibiotics, or was the effect specific to doxycycline?
8. Fig 3 - 8. Eliminate, as noted above.
9. Fig 9, Consider eliminating panels A and B, keeping the lung study. A: I am unaware of any situation in which *Galleria* infection is more predictive of human efficacy than mouse infection. B: The drug was administered directly to the site of infection (peritoneum), so this study is of little value.
10. Fig 9, cytokine study (panels G-I). This requires more explanation. Were cytokines determined in plasma from the same mice as in the efficacy study? If not, please describe the experiment (route of infection, route of drug administration, timing of sample collection, etc). Note that reduced cytokine reduction is consistent with smaller numbers of bacteria; these observations do not necessarily suggest an anti-inflammatory effect.
11. Move fig S6 into the main text, merging it with fig 9
12. Eliminate figs S1 - S5
13. Technical details: Baicalein was prepared as a DMSO solution (line 269). What was the concentration of DMSO in MIC determinations? Cytotoxicity determinations presumably included serum in cell culture medium. What is the effect of added serum on the antibacterial activity of baicalein, doxycycline, or the combination?
14. Terminology:
 - a. The term synergy should be avoided except where the experimental design allows one to distinguish between synergy and additivity.
 - b. The term "the cell membrane" is used several times (line 103, lines 106-107, lines 210-211). This term makes no sense and should be replaced by a phrase that is consistent with Gram-negative bacteria possessing two membranes. For example, Lines 106-107, "In bacteria, the cell membrane is the first line of defense against antibiotics (21)." Suggest "In bacteria, the cell envelope is the first..." Also, check the reference. Ref 20 (Hurdle) might be more appropriate.
15. The writing is mostly clear and grammatical but does include some errors. Suggest editing for English.
16. Minor point: Reviewing would be made easier by labeling the figures (Fig 1, fig 2, etc) and by adding the legends at the bottom of each figure (similar to the supplemental material).
17. Reference list: Inclusion of doi numbers is much appreciated!

Staff Comments:

Preparing Revision Guidelines

Please return the manuscript within 60 days; if you cannot complete the modification within this time period, please contact me. If you do not wish to modify the manuscript and prefer to submit it to another journal, please notify me of your decision immediately so that the manuscript may be formally withdrawn from consideration by Microbiology Spectrum.

Comments of Reviewer 1

General comments: *The manuscript by Want et al. examines the effect of baicalein on bacteria upon co-treatment with doxycycline. It is overall a thoughtful treatment of an important question. Baicalein is a flavonoid from a plant root that has been used in traditional Chinese medicine for treating bacterial infections. The authors show that baicalein potentiates multiple antibiotics in broth against Gram-negative bacteria and then focus in on doxycycline. Through a series of assays, the authors show that co-treatment with dox and baicalein kills bacteria but does not kill or lyse mammalian cells. Baicalein damages A. baumannii outer and inner membranes and appears to specifically compete with magnesium to bind lipid A. The compound also likely interacts with bacterial lipids, especially PG. Transcriptomics comparing dox to dox + baicalein revealed many gene changes. Co-treatment also disrupted biofilms, reduced host responses to LPS, reduced mortality in wax-worms and reduced tissue colonization and inflammatory cytokines in mice. These data indicate that baicalein works through multiple mechanisms both at the level of the bacterium and on the host.*

Overall response to Reviewer 1: Thank you for reviewing our manuscript and providing some kind comments. In the revised version, the new text is highlighted in yellow, and the underline indicates the revised text to identify better the changes made to the previous version. Our point-to-point response is provided below in red text.

In what follows, we would like to answer the questions you mentioned and give a detailed account of the changes made to the original manuscript.

Comment 1: *For the assays with florescent or luminescence probes, there need to be controls in which compound is incubated with the probe in the absence of cells at the concentrations of probe and compound used in the corresponding experiment. This is*

necessary to establish whether a compound on its own alters signal from the probe. These controls, or descriptions of the controls, were not found in the methods for Figure 7, for instance.

Response: Since we did not express it clearly, we are sorry for your misunderstanding. Actually, in fluorescence or luminescence assay, the corresponding concentration of drugs (without cells) were incubated with probes as blank control, and the relative fluorescence intensity was calculated using the analyte value minus the blank controls. To be more explicit and to follow the reviewer's concerns, we have added a brief description as follows: "In fluorescence or luminescence assay, the corresponding concentration of drugs (without cells) were incubated with probes as blank control, and the relative fluorescence intensity was calculated using the analyte value minus the blank controls." (Page 16, line 347-350)

Comment 2: The transcriptomics are difficult to interpret without a baicalein-only sample.

Response: Thanks for your valuable counsel. The transcriptomic datasets of baicalein alone have been included in the revised manuscript (Figure 6).

Moreover, we have added overall and commonality description of new transcriptome-based data: "Compared with the doxycycline alone, the combination treatment contained 552 upregulated genes and 313 downregulated genes, while the combination group showed 704 genes upregulated and 505 genes downregulated when compared to the baicalein alone (Figure 6A). As revealed by KEGG analysis, the common differentially expressed genes were associated with microbial metabolism, protein export, oxidative phosphorylation, TCA cycle, and ABC transporters (Figure 6B)." (Page 8, line 153-158).

Because our focus herein is on the mechanism of action of the baicalein combined with doxycycline against doxycycline-resistant gram-negative bacteria, in the following research, we conducted an analysis focused on gene transcription levels of the combination compared to the doxycycline alone.

Comment 3: *Figure 8 A and B - It would be more transparent to display Figure 8 A and B as bar graphs instead of line graphs, as the data are derived from different samples. Also, no error bars are visible, and the legend does not mention that there are error bars that are too small to be seen.*

Response: Thank you for pointing this out. We have changed Figure 8 A and B to bar graphs with error bars, and the bar graphs show individual data points (Figure 8).

Comment 4: *Sup Fig 1 E-H. Please put how long after treatment in the legend.*

Response: We are grateful for the suggestion and have added the specific treatment time in the Figure legend. "(E-H) Morphological changes of *K. Pneumoniae* KP1 treated with PBS (E); sub-MIC of doxycycline (64 µg/ml) (F); or sub-MIC of baicalein (125 µg/ml) (G); or their combination for 1 h (H) visualized with SEM." (Supplementary Figure 2).

"(E-H) Morphological changes of *A. baumannii* AB145 treated with PBS (E), sub-MIC of doxycycline (64 µg/mL) (F), sub-MIC of baicalein (125 µg/mL) (G), or their combination for 1 h (H), visualized using SEM." (Page 38, line 780-782).

Comment 5: *Ideally, bar graphs show individual data points so that readers can see data spreads.*

Response: Thanks for your kind suggestions, which are valuable for improving the manuscript's accuracy. We have changed all bar graphs to show individual data

points.

Comments of Reviewer 2

General comments: *This manuscript seeks to characterize the antibacterial activity of a natural product, baicalein, with particular focus on its ability to potentiate doxycycline activity in tetracycline-resistant Gram-negative bacteria. Baicalein is one of the active components in certain extracts used in Chinese traditional medicine that are thought to enhance the effectiveness of antibiotics in patients. As an isolated compound, it has been studied previously by others in order to evaluate its utility in various therapeutic indications, not all infectious disease. Its synergy with tetracyclines was studied in *S aureus* and was initially thought to result from inhibition of the TetK pump; it was then realized that baicalein could potentiate tetracycline in MRSA strains lacking TetK and that it could potentiate a variety of antibiotic classes in various bacterial species.*

*Li and colleagues undertook the current study with the idea that baicalein might restore the utility of doxycycline for treatment of Gram-negative infections. They found using checkerboard MICs that baicalein is technically synergistic with multiple antibiotics. The greatest effect is seen for doxycycline, where the MIC is reduced by up to 32 fold in some MDR strains of *A baumannii*. However, the resulting MIC is generally well above the clinical breakpoint. Moreover, these effects require fairly high concentrations of baicalein, 31.25 or 62.5 ug/ml. Although baicalein appears to be fairly well tolerated in mice, it seems unlikely that it will be feasible to reach these concentrations in plasma of patients treated with the proposed combination.*

It is thus extremely unlikely that baicalein could be used clinically as an antibiotic partner to treat infections. Despite the very modest antibacterial activity of balicalein (MIC 250 ug/ml), one might feel it is worth studying, perhaps leading to novel approaches for designing agents with greater activity as standalone antibiotics or

potentiators.

Much of the manuscript describes the attempt to determine the mechanism of action of baicalein and/or the mechanism by which it potentiates other antibacterial agents. Unfortunately, most of the mechanistic studies are not interpretable. In some studies, bacteria are treated with drugs for an hour or longer, with 4 hr for the gene expression studies. This is inappropriate, as the direct effects of antibacterial agents on bacterial physiology are generally apparent within 20-30 minutes. With longer times, the observed effects are indirect and shed no light on the molecular mechanism of the drug. Several experiments compared a subinhibitory treatment (single drug) to an inhibitory treatment (two drugs). It is likely that the effects seen are simply the effect of stress, growth inhibition or cell death. Any other inhibitory treatment might produce the same results.

I recommend shortening the manuscript substantially, focusing on the stronger parts of the work: in-vitro and in-vivo microbiology. I recommend delaying publication of the mechanistic studies until they have been conducted more carefully. These experiments should have included controls - i.e., comparator antibiotics with known mechanisms, and compounds previously characterized as OM disrupting agents, ionophores, or pore-forming agents. Bacterial viability should be monitored. Experimental conditions such as incubation times or concentration of reagents should be justified unless well supported by literature that is cited.

Overall response to Reviewer 2: Thank you for spending time reviewing our manuscript and providing us with a list of constructive comments. We understand that you have some significant concerns regarding that although baicalein combined with doxycycline, where the MIC of doxycycline substantially decreases in some

MDR strains of *A baumannii*, but some of the resulting MICs are well above the clinical breakpoint. According to the 2021 guidelines, doxycycline sensitivity is defined as $\text{MIC} \leq 4 \mu\text{g/mL}$; intermediate is defined as $4 < \text{MIC} < 16 \mu\text{g/mL}$; resistance is defined as $\text{MIC} \geq 16 \mu\text{g/mL}$ (1). After careful analysis, it was found that 98/245 (40%) *A baumannii* isolates were high-MIC of doxycycline-resistant strains ($\geq 128 \mu\text{g/mL}$), and their resistance phenotype changed from resistant to sensitive for doxycycline after the baicalein treatment. 90/245 (36.73%) *A baumannii* isolates were lower-MIC of doxycycline-resistant strains ($16 \leq \text{MIC} \leq 64 \mu\text{g/mL}$), and their resistance phenotype changed from resistant to intermediate or sensitivity for doxycycline after the baicalein treatment. In addition, 21/245 (8.57%) *A baumannii* isolates were high-MIC of doxycycline-resistant strains ($\geq 128 \mu\text{g/mL}$) and they are all pandrug-resistant strains, the synergistic MIC of which is above the clinical breakpoint. This could be due to the pandrug-resistant and high-MIC of doxycycline strains with a robust antibiotic-resistance gene system that limits baicalein and doxycycline's combined effects.

An excessive concentration of baicalein is used in the combination, which causes it challenging to reach effective drug concentrations in the plasma of patients. Nevertheless, we recognized this limitation should be mentioned in the paper, so we added the following sentence "This research reveals that baicalein could be considered a lead compound that merits further optimization and development as a candidate adjuvant that helps combat antibiotic resistance." (Page 2, line 35-37).

"Due to its low cytotoxicity and resistance, the baicalein/doxycycline combination provides a valuable clinical reference for selecting more effective therapeutic strategies for treating multidrug-resistant gram-negative clinical isolate infections."

(Page 2, line 42-45).

"Nonetheless, some limitations must be addressed in the future. Our discovery that baicalein can significantly reduce the MIC of doxycycline in gram-negative isolates; however, an excessive concentration of baicalein is used in the combination, which causes it challenging to reach effective drug concentrations in the plasma of patients. Future research should focus on optimizing baicalein's pharmacologic properties, which have more significant antimicrobial activity and higher bioavailability as standalone antibiotics or potentiators." (Page 13, line 277-283).

We thank the reviewer for the fascinating comments. We are considering longer drug applications to bacteria that inhibit the activity of bacteria caused by the stress and not the drugs. We referred to previous studies in this section and re-examined the mechanistic studies with drugs, including shortening the treatment period to 30 mins, adjusting drug concentrations, adding positive control, et al. Modified throughout the text according to the comment.

Cell inner and outer membranes integrity assay. Fresh bacterial culture ($OD_{600} = 0.5$) with the indicated concentrations of baicalein or PB (polymyxins B, 16 $\mu\text{g}/\text{mL}$) at room temperature incubated for 30 mins. Next, the bacterial suspensions were mixed with 10 nmol/L of propidium iodide (PI; Solarbio, Beijing, China), which was performed to detect the permeabilization of the inner membrane, and co-incubated for 30 mins protected from the light. The fluorescence was detected at 615/535 nm for emission/excitation wavelengths (2). In fluorescence or luminescence assay, the corresponding concentration of drugs (without cells) were incubated with probes as blank control, and the relative fluorescence intensity was calculated using the analyte value minus the blank controls.

For the outer membrane permeability assay, 10 $\mu\text{mol/L}$ of 1-N-phenyl-naphthylamine (NPN; Solarbio, Beijing, China) was performed following the same procedure, colistin (4 $\mu\text{g/mL}$) as a positive control, and the fluorescence intensity was measured at 420/350 nm for emission/excitation wavelengths (3)." (Page 16, line 341-354).

"Extracellular β -galactosidase measurement. Different concentrations of baicalein or PB (polymyxins B, 16 $\mu\text{g/mL}$) were co-incubated with bacterium solution ($\text{OD}_{600} = 0.5$) for 30 mins (37°C). Then, 190 μL of supernatants (12,000 g, 15 min, 4°C) were incubated with 2-nitrophenyl- β -d-galactopyranoside (3 mmol/L, ONPG, Sigma) for 30 min, and the OD_{420} was detected (4)." (Page 16, line 355-359).

"Measurement of ATP content. Bacterium solution ($\text{OD}_{600} = 0.5$) was co-incubated with the indicated concentrations of drugs for 30 mins." (Page 17, line 360-361).

"Scanning electron microscopy (SEM). Doxycycline (64 $\mu\text{g/mL}$), baicalein (125 $\mu\text{g/mL}$), or their combination was used to treat bacterium solution ($\text{OD}_{600} = 0.5$) for 1 h, followed by fixed overnight using 2.5% glutaraldehyde at 4°C. After thoroughly rinsing thrice using PBS, bacteria were subjected to 10 min gradual dehydration using ethanol (30%, 50%, 70%, 80%, 90%, 95%) and 10 min dehydration using 100% ethanol thrice. The samples were centrifuged at 4,000 g for 10 min to remove fixatives, followed by resuspension of bacterial pellets in PBS (1 mL). After processing, each sample was dried using a critical point dryer, coated with gold-palladium using an ion sprayer (5) and then observed with SEM (GeminiSEM 300)." (Page 17, line 365-373).

"Lipopolysaccharide (LPS), cationic ion, and phospholipid assays. Bacterial LPS was extracted using a kit from iNtRON Biotech (Kyungki-Do, Korea). Next, LPS, bai

calein, doxycycline, and bacterial suspension (5×10^8 CFUs/mL) were added into a 96-well plate, followed by 18 h incubation at 37°C. The checkerboard microdilution assay was applied to assess how LPS (0-128 μ g/mL) affected baicalein's antibacterial effect. MIC assays were used to determine the impact of different cationic ions (50 μ g/mL), which included KCl, CaCl₂, MgCl₂, CuSO₄, FeCl₃, NaCl, MgCl₂, and ZnSO₄ (Aladdin, Shanghai, China), on baicalein's antibacterial effect on *A. baumannii* or *K. pneumoniae*. Phosphatidylglycerol (PG; Sigma-Aldrich, 841188P, $\geq 99\%$), phosphatidylcholine (PC; Sigma-Aldrich, catalog no. 840051P, $\geq 99\%$), cardiolipin (CL; Sigma-Aldrich, 841199P, $\geq 99\%$), and phosphatidylethanolamine (PE; Sigma-Aldrich, 840027P, $\geq 99\%$) were subjected to methanol dissolution. The checkerboard microdilution assay was used to assess how different phospholipid (0–128 μ g/mL) concentrations affected baicalein's antibacterial effect in MHB." (Page 17, line 374-388).

TCA cycle measurement. Doxycycline (0–512 μ g/mL) with/without baicalein (125 μ g/mL) was added to bacterial suspensions ($OD_{600} = 0.5$). Bacterial pellets were washed with PBS after **1 h** of incubation, followed by resuspension in 200 μ L pre-chilled extraction buffer. The lysates were centrifuged at 12,000 g for 10 min at 4°C, and supernatants were analyzed by adopting NAD⁺/NADH Assay Kit with WST-8 (Beyotime, China) **(6)**." (Page 19, line 409-414).

Total ROS, superoxide dismutase (SOD), and hydrogen peroxide (H₂O₂) measurement. The bacterial suspension ($OD_{600} = 0.5$) was treated with 0.01 mM 2',7'-dichlorodihydrofluorescein diacetate (DCFH-DA; Invitrogen) to measure ROS levels for 30 mins at 37°C. The bacterial suspension (190 μ L) mixed with 10 μ L the indicated concentrations of baicalein were added to 96 well plates incubated for **30 mins**. The fluorescence intensity was detected at excitation/emission wavelengths of 488/525 nm **(7)**."

(Page 19, line 415-421).

"Next, the bacterial solution was mixed with varying concentrations of baicalein, baicalin combined with doxycycline, or **benzyl alcohol (50 mmol/L)** for 30 mins." (Page 19, line 427-429).

"Then 10 μ L of varying drugs, glucose (2.5×10^{-5} M) or **Melittin (16 μ g/mL)**, were supplemented and incubated for **30 mins**, and the fluorescence values of PMF and dissipated bacterial $\Delta\Psi$ were determined by excitation/emission at 500/522 nm and 622/670 nm, respectively (2)." (Page 20, line 435-438).

The concentration and exposure time used in the transcriptome sequencing experiment was determined two-fold. On the one hand, we referred to previous studies (3, 8, 9). On the other hand, the time-kill curve indicated that after 8 hours of incubation, growth characteristics of bacterial populations tended to be stabilized, 4 h as midpoint time, and the bacterial loads of combined treatment were found to be approximately 3 orders of magnitude lower than other groups. We have added the information required as explained above. "*A. baumannii* AB145 was grown in MHB till the early exponential phase; next, bacterial cells were treated with 64 μ g/mL doxycycline, 125 μ g/mL baicalein alone, or their combination for 4 h (9)." (Page 18, line 398-400).

Thank the reviewer for the constructive comments and suggestions. In the revised version, the new text is highlighted in yellow, and the underline indicates the revised text to identify better the changes made to the previous version. Our point-to-point response is provided below in red text. We hope these changes improve the clarity and accuracy of the presentation.

In what follows, we would like to answer the questions you mentioned and give a

detailed account of the changes made to the original manuscript.

Comment 1: *Move tables S1 and S2 into the main text.*

Response: Thank you for this suggestion. We merged supplementary Table 1 and supplementary Table 2 into Table 2.

Comment 2: *Fig 1B. Please explain the spider-web plots. A table showing FICI for each antibiotic would be much easier to read and would allow the MIC of each antibiotic alone to be included. Why was this experiment done with strain AB43 Δ crispr-cas? What are the results for the drug-sensitive parent AB43?*

Response: We gratefully appreciate for your valuable comment. Figure 1B was switched to Table 1.

" For our previously constructed XDR (extensively drug-resistant) *A. baumannii* AB43 Δ crispr-cas (10), a checkerboard assay showed that baicalein could act as a synergistic bacteriostatic effect (FICI \leq 0.5) with most of the conventional antibiotics (10/14, 71.43%), which included doxycycline (FICI = 0.1875) (Table 1). A similar phenomenon was found in the sensitive strain AB43 (Table S1)." (Page 6, line 95-100).

Comment 3: *Fig 1C and Table S2 show the same data for the same ten isolates of A baumannii, but in a different order. Please present both in the same order, perhaps numerical by strain ID.*

Response: Thank you for your reminder. We performed sequential adjustments based on strain ID. (Figure 1B)

Comment 4: *Fig 1D requires some explanation. What does "inhibition rate" mean? How was the experiment done?*

Response: Thank you for the suggestion. "According to the CLSI 2021 guidelines, no visible growth of bacteria was considered growth inhibition when bacteria co-incubated with a baicalein and doxycycline compound for 18 hours at 37°C." (Page 36, line 746-749). The "inhibition rate" means the number of strains whose growth is inhibited by the drug combinations as a percentage of 245.

Comment 5: Fig 2. Eliminate panel B. The untreated bacteria are growing and are clearly not in stationary phase.

Response: Thank you for pointing this out. We have deleted panel B of Figure 2B.

Comment 6: Fig 2 C-F, hemolysis and cytotoxicity. Low levels of both hemolysis and cytotoxicity are seen at very high concentrations of doxycycline (256 or 512 ug/ml), except in the presence of extremely high concentrations of baicalein (250 ug/ml). Lines 638-641 interpret these observations as showing baicalein protecting mammalian cells from detrimental effects of doxycycline. Is it generally thought that cytotoxicity of doxycycline is clinically relevant? If not, I suggest rewriting this section to focus on the lack of cytotoxicity and hemolysis for baicalein at concentrations that are likely to be achieved in vivo - both alone and in the presence of doxycycline.

Response: We are grateful for the suggestion. To be more precise and per the reviewer's concerns, we have added a brief description: "(C-F) Safety evaluation of baicalein. Different concentrations of baicalein (2-250 µg/mL) had no significant effects on the hemolytic toxicity of the red blood cells of sheep (C) and cytotoxicity in A549, THP-1, and RAW264.7 cells (D-F) caused by doxycycline (0-128 µg/mL)" (Page 37, line 762-765).

Comment 7: Fig 2G and G, serial passage studies. At the start of the experiment, the

MIC of doxycycline is 128 ug/ml for both AB145 and KP1. The plot shows the MICs increasing by 32 or 64 fold during passage in doxycycline unless baicalein was included. That would mean the final MICs are 4 or 8 milligrams/milliliter - is that correct? Suggest plotting actual MICs on the Y axis, rather than fold increase. Were these extremely high MICs still seen after the final culture had been passaged on drug-free agar? Did the resistant isolates have elevated MICs for other antibiotics, or was the effect specific to doxycycline?

Response: Considering the Reviewer's suggestion, we have plotted actual MICs on the Y axis of Figure 2G and H.

We are very sorry about the negligence which has confused reviewers. "**Resistance development studies.** Bacterium solution (0.5×10^6 CFUs/mL) contains doxycycline ($0.25 \times \text{MIC}$) alone or in combination with baicalein ($62.5 \mu\text{g/mL}$) incubated at 220 rpm for 24 hours (37°C). Subsequently, the visible growth ($\text{OD}_{600} \geq 0.3$) strains were passaged on antibiotic-free MHA plates for 24 h, and the corresponding MICs of doxycycline were detected by the two-fold serial dilution method described above. Simultaneously, the bacterium solution ($\text{OD}_{600} = 0.5$) was diluted 1:100 with MHB and incubated in the presence of doxycycline ($1/4 \text{ MIC}$) or combination with baicalein ($62.5 \mu\text{g/mL}$) for the next generation. This step was performed repeatedly for 30 days; then, the fold increase of the doxycycline MIC compared with the original MIC was determined (11)." (Page 15, line 329-338).

As the reviewer points out, it is an excellent suggestion to elevate MICs for other antibiotics in doxycycline-resistant mutants. "Antibiotic resistance is a significant challenge in the treatment of bacterial infections. Serial passaging for 31 days in the presence of doxycycline increased MIC for *A. baumannii* AB145 and *K. pneumoniae*

KP1 by 32 and 64 folds, respectively. In contrast, the combination treatment did not allow resistance to develop (Figure 2G-H). It is worth noting that the per-generation doxycycline-resistant mutants exhibited cross-resistance (12) to multiple different classes of antibiotics, and the expression levels of *tetA* were significantly increased in some high-level doxycycline-resistant mutants (Figure S1, Table S2). Furthermore, all of the doxycycline-resistant mutants had multiple nonsynonymous mutations in the *tetA* genes (Table S3). Baicalein was found to be effective in inhibiting doxycycline resistance in *tetA*-positive MDR *A. baumannii* and *K. pneumoniae* KP1" (Page 6, line 108-118).

Comment 8: Fig 3 - 8. Eliminate, as noted above.

Response: Thank you for the suggestion. To better understand the underlying mechanisms of baicalin combined with doxycycline, we referred to previous studies in this section. We re-examined the mechanistic studies with drugs, including shortening the treatment period to 30 mins, adjusting drug concentrations, adding positive control et al. The results trends are consistent with previous studies. (Figure 3-8)

Comment 9: Fig 9, Consider eliminating panels A and B, keeping the lung study. A: I am unaware of any situation in which *Galleria* infection is more predictive of human efficacy than mouse infection. B: The drug was administered directly to the site of infection (peritoneum), so this study is of little value.

Response: You have raised an important point here. However, we believe that the *Galleria mellonella* infection and mouse peritonitis sepsis model would be more appropriate in this study because the determination of survivorship in these infection models remains a primary model for evaluation of the efficacy and safety of

antimicrobial agents (3, 6, 8, 9, 11, 13-16).

Comment 10: *Fig 9, cytokine study (panels G-I). This requires more explanation. Were cytokines determined in plasma from the same mice as in the efficacy study? If not, please describe the experiment (route of infection, route of drug administration, timing of sample collection, etc). Note that reduced cytokine reduction is consistent with smaller numbers of bacteria; these observations do not necessarily suggest an anti-inflammatory effect.*

Response: Thank you for your significant reminding. We have rewritten this part according to the reviewer's suggestion. "**Neutropenic mouse lung infection model.** Cyclophosphamide (150 and 100 mg/kg) was administered to female BALB/c mice (n = 8/group) on days 4 and 1 before infection for neutropenic induction. After that, the lungs of mice were injected with 100 μ L of bacterial suspension (1.0×10^6 CFUs/mouse) (17). At 2-h post-infection, the mice were administered an intraperitoneal injection of PBS, doxycycline (50 mg/kg), baicalein (50 mg/kg), or their combination (50 + 50 mg/kg). After 48 h, each mouse was euthanized by cervical dislocation. Peripheral blood serum samples of mice were separated and stored at -80°C for cytokine detection. Mouse organs were dissected under septic conditions, followed by homogenization, serial dilution, and inoculation in MHA for CFU counting. Meanwhile, those mice's left upper lung tissues were excised aseptically for hematoxylin and eosin (HE) staining." (Page 22, line 480-491).

We are grateful for the suggestion. To be more transparent and following the reviewer's concerns, we have added the HE staining of lung tissue from different treated mice and a brief description as follows: "Moreover, in the combined treatment group, the cytokine serum levels of four inflammatory cytokines (IL-1 β , IFN- γ , TNF-

α , IL-8) were markedly decreased at 48 hours after pulmonary infection compared with other groups. confirmed by the alleviated pulmonary pathology (Figure S6)."

(Page 10, line 210-213).

Comment 11: *Move fig S6 into the main text, merging it with fig 9.*

Response: As the reviewer suggested, it is a good idea, and we have merged Figure S6 with Figure 9.

Comment 12: *Eliminate figs S1 - S5.*

Response: Thank you for the suggestion. To better understand the underlying mechanisms of baicalin combined with doxycycline, we referred to previous studies in this section. We re-examined the mechanistic studies with drugs, including shortening the treatment period to 30 mins, adjusting drug concentrations, adding positive control et al. The results trends are consistent with previous studies. (Figure S2-S5)

Comment 13: *Technical details: Baicalein was prepared as a DMSO solution (line 269). What was the concentration of DMSO in MIC determinations? Cytotoxicity determinations presumably included serum in cell culture medium. What is the effect of added serum on the antibacterial activity of baicalein, doxycycline, or the combination?*

Response: We understand the reviewer's concern. The solubility of baicalin in DMSO is 54 mg/mL (199.82 mM). "According to the Clinical and Laboratory Standards Institute (CLSI, USA), the baicalein was dissolved in dimethyl sulfoxide (DMSO) with a final concentration of less than 1%." (Page 14, line 295-297).

Thank you for your rigorous consideration. "16HBE and THP-1 cells were incubated

in RPMI-1640 medium (Gibco) supplemented with 10% heat-inactivated FBS (Invitrogen) and 50 μ M β -mercaptoethanol. RAW264.7 and A549 cells were passaged in DMEM (Gibco) with 10% heat-inactivated FBS (Invitrogen)." (Page 14, line 290-294). Through heat inactivation (56°C for 30 min) to remove the serum complement does not affect the MIC of baicalein, doxycycline, or their combination.

Comment 14: Terminology: a. The term synergy should be avoided except where the experimental design allows one to distinguish between synergy and additivity.

b. The term "the cell membrane" is used several times (line 103, lines 106-107, lines 210-211). This term makes no sense and should be replaced by a phrase that is consistent with Gram-negative bacteria possessing two membranes. For example, Lines 106-107, "In bacteria, the cell membrane is the first line of defense against antibiotics (21)." Suggest "In bacteria, the cell envelope is the first..." Also, check the reference. Ref 20 (Hurdle) might be more appropriate.

Response: Thank you for your significant reminding. We have modified the sentence according to the previous comment. "The calculated FICIs were defined as synergistic ($\text{FIC} \leq 0.5$), additive ($0.5 < \text{FIC} < 1$), indifferent ($1 \leq \text{FIC} < 4.0$), or antagonistic ($\text{FIC} \geq 4.0$)." (Page 14, line 308-310).

We have changed "The exogenous addition of the corresponding bacterial strain source of LPS reduces the antibacterial activity of baicalin dose-dependently, and high concentrations of LPS (128 μ g/mL) diminished the synergistic effect between baicalein and doxycycline (Figure 4A-B)." to "The exogenous addition of the corresponding bacterial strain source of LPS reduces the antibacterial activity of baicalin dose-dependently, and high concentrations of LPS (128 μ g/mL) diminished the antibacterial effects of baicalein combined doxycycline (Figure 4A-B)." (Page 7,

line 131-134).

We have changed "We found that the exogenous addition of K^+ , Ca^{2+} , Mg^{2+} , Cu^{2+} , and Zn^{2+} inhibited the activity of baicalein, and Mg^{2+} strongly inhibited the synergistic antibacterial effects, while Fe^{3+} , Mn^{2+} , and Na^+ had no significant impact on the action (Figure 4C-E)." to "We found that the exogenous addition of K^+ , Ca^{2+} , Mg^{2+} , Cu^{2+} , and Zn^{2+} inhibited the activity of baicalein, and Mg^{2+} strongly inhibited the antibacterial effects of baicalein combined doxycycline, while Fe^{3+} , Mn^{2+} , and Na^+ had no significant impact on the action (Figure 4C-E)." (Page 7, line 136-139).

We have changed "The MICs of baicalein are higher with the exogenous addition of phospholipids, and PG blocks the synergy between baicalein and antibiotics (Figure 5). These data suggest that baicalein-mediated membrane perturbation is vital for its synergistic effect with doxycycline." to "The MICs of baicalein are higher with the exogenous addition of phospholipids, and PG blocks the antibacterial effects of baicalein combined with antibiotics (Figure 5). These data suggest that baicalein-mediated membrane perturbation is vital to promote the antibacterial efficacy of doxycycline." (Page 8, line 145-148).

We have revised the text to address your concerns and hope it is more evident. We have made the change. "Baicalein disrupts the cell membrane of gram-negative bacteria." to "Baicalein disrupts the inner and outer membranes of gram-negative bacteria." (Page 7, line 119-120).

"In bacteria, the cell membrane is the first line of defense against antibiotics (18)." to "In bacteria, the cell envelope is the first line of defense against antibiotics (19)." (Page 7, line 122-123).

"The SEM images revealed that after the combination treatment, the cell surface was

depressed, shrunk, collapsed, and lysed compared with no treatment, indicating that baicalein's effect on bacteria was possibly associated with the rapidly disrupted cell wall and cell membrane." to "The SEM images revealed that after the combination treatment, the cell surface was depressed, shrunk, collapsed, and lysed compared with no treatment, indicating that baicalein's effect on bacteria was possibly associated with the rapidly disrupted cell wall and the cell's inner and outer membranes." (Page 7, line 125-129).

"The bacterial cell membrane is fluid, which is critical for bacterial proliferation and survival (20)." to "The bacterial outer membrane is fluid, critical for bacterial proliferation and survival (20)." (Page 9, line 176-177).

"Baicalein exhibited synergistic antibacterial effects by disrupting the cell membrane of gram-negative bacteria." to "Baicalein exhibited synergistic antibacterial effects by disrupting gram-negative bacteria's inner and outer membranes." (Page 11, line 231-232).

"In conclusion, our findings first indicated that baicalein resensitizes *tetA*-positive MDR gram-negative pathogens to doxycycline by direct engagement with phospholipids and lipopolysaccharides, thereby disturbing the permeability of the gram-negative bacterial cell membrane." to "In conclusion, our findings first indicated that baicalein sensitizes *tetA*-positive MDR gram-negative pathogens to doxycycline by direct engagement with phospholipids and lipopolysaccharides, thereby disturbing the permeability of the gram-negative bacterial cell inner and outer membranes." (Page 13, line 271-274).

We feel sorry for the inconvenience brought to the reviewer. We have adjusted the references. "In bacteria, the cell envelope is the first line of defense against antibiotics

(19)." (Page 7, line 122-123).

Comment 15: *The writing is mostly clear and grammatical but does include some errors. Suggest editing for English.*

Response: This manuscript has been revised extensively according to the reviewers' constructive suggestions. In addition, the expression of the manuscript has been improved with the help of a native English speaker. The certificate of English language editing has been uploaded to the system along with the revised manuscript.

Comment 16: *Minor point: Reviewing would be made easier by labeling the figures (Fig 1, fig 2, etc) and by adding the legends at the bottom of each figure (similar to the supplemental material).*

Response: Thank you for pointing this out. We have made corrections.

Comment 17: *Reference list: Inclusion of doi numbers is much appreciated!*

Response: Thank you for pointing this out. We have corrected it according to the reviewer's comments.

We tried our best to improve the manuscript and made some changes in the manuscript. These changes will not influence the content and framework of the paper. And here, we did not list the changes but marked them in red in the revised paper.

We appreciate for Editors/Reviewers' warm work earnestly and hope that the correction will meet with approval.

We look forward to hearing from you regarding our submission and responding to any further questions and comments you may have.

Once again, thank you very much for your comments and suggestions.

Sincerely,

Corresponding author:

Name: Guocai Li

E-mail: gcli@yzu.edu.cn

References

1. (CLSI). CaLSI. March 23, 2021. . M100 Performance Standards for Antimicrobial Susceptibility Testing, 31th ed. <https://www.clsi.org/standards/products/microbiology/documents/m100/>. Accessed
2. She P, Li Z, Li Y, Liu S, Li L, Yang Y, Zhou L, Wu Y. 2022. Pixantrone Sensitizes Gram-Negative Pathogens to Rifampin. *Microbiol Spectr* 10:e0211422.
3. Song M, Liu Y, Huang X, Ding S, Wang Y, Shen J, Zhu K. 2020. A broad-spectrum antibiotic adjuvant reverses multidrug-resistant Gram-negative pathogens. *Nat Microbiol* 5:1040-1050.
4. Kang DD, Park J, Park Y. 2022. Therapeutic Potential of Antimicrobial Peptide PN5 against Multidrug-Resistant *E. coli* and Anti-Inflammatory Activity in a Septic Mouse Model. *Microbiol Spectr* 10:e0149422.
5. Fujimura S, Sato T, Mikami T, Kikuchi T, Gomi K, Watanabe A. 2008.

- Combined efficacy of clarithromycin plus cefazolin or vancomycin against *Staphylococcus aureus* biofilms formed on titanium medical devices. *Int J Antimicrob Agents* 32:481-4.
6. Liu Y, Yang K, Jia Y, Shi J, Tong Z, Wang Z. 2021. Thymine Sensitizes Gram-Negative Pathogens to Antibiotic Killing. *Front Microbiol* 12:622798.
 7. Carter WO, Narayanan PK, Robinson JP. 1994. Intracellular hydrogen peroxide and superoxide anion detection in endothelial cells. *J Leukoc Biol* 55:253-8.
 8. Liu Y, Jia Y, Yang K, Li R, Xiao X, Zhu K, Wang Z. 2020. Metformin Restores Tetracyclines Susceptibility against Multidrug Resistant Bacteria. *Adv Sci (Weinh)* 7:1902227.
 9. Liu Y, Jia Y, Yang K, Tong Z, Shi J, Li R, Xiao X, Ren W, Hardeland R, Reiter RJ, Wang Z. 2020. Melatonin overcomes MCR-mediated colistin resistance in Gram-negative pathogens. *Theranostics* 10:10697-10711.
 10. Wang Y, Yang J, Sun X, Li M, Zhang P, Zhu Z, Jiao H, Guo T, Li G. 2022. CRISPR-Cas in *Acinetobacter baumannii* Contributes to Antibiotic Susceptibility by Targeting Endogenous *AbaI*. *Microbiol Spectr* doi:10.1128/spectrum.00829-22:e0082922.
 11. Maisuria VB, Okshevsky M, Déziel E, Tufenkji N. 2019. Proanthocyanidin Interferes with Intrinsic Antibiotic Resistance Mechanisms of Gram-Negative Bacteria. *Adv Sci (Weinh)* 6:1802333.
 12. Lázár V, Nagy I, Spohn R, Csörgő B, Györkei Á, Nyerges Á, Horváth B, Vörös A, Busa-Fekete R, Hrtyan M, Bogos B, Méhi O, Fekete G, Szappanos B, Kégl B, Papp B, Pál C. 2014. Genome-wide analysis captures the determinants of the antibiotic cross-resistance interaction network. *Nat*

Commun 5:4352.

13. Chen S, Liu D, Zhang Q, Guo P, Ding S, Shen J, Zhu K, Lin W. 2021. A Marine Antibiotic Kills Multidrug-Resistant Bacteria without Detectable High-Level Resistance. *ACS Infect Dis* 7:884-893.
14. De Oliveira DMP, Bohlmann L, Conroy T, Jen FE, Everest-Dass A, Hansford KA, Bolisetti R, El-Deeb IM, Forde BM, Phan MD, Lacey JA, Tan A, Rivera-Hernandez T, Brouwer S, Keller N, Kidd TJ, Cork AJ, Bauer MJ, Cook GM, Davies MR, Beatson SA, Paterson DL, McEwan AG, Li J, Schembri MA, Blaskovich MAT, Jennings MP, McDevitt CA, von Itzstein M, Walker MJ. 2020. Repurposing a neurodegenerative disease drug to treat Gram-negative antibiotic-resistant bacterial sepsis. *Sci Transl Med* 12.
15. Feng X, Liu S, Wang Y, Zhang Y, Sun L, Li H, Wang C, Liu Y, Cao B. 2021. Synergistic Activity of Colistin Combined With Auranofin Against Colistin-Resistant Gram-Negative Bacteria. *Front Microbiol* 12:676414.
16. Sun H, Zhang Q, Wang R, Wang H, Wong YT, Wang M, Hao Q, Yan A, Kao RY, Ho PL, Li H. 2020. Resensitizing carbapenem- and colistin-resistant bacteria to antibiotics using auranofin. *Nat Commun* 11:5263.
17. Lesic B, Starkey M, He J, Hazan R, Rahme LG. 2009. Quorum sensing differentially regulates *Pseudomonas aeruginosa* type VI secretion locus I and homologous loci II and III, which are required for pathogenesis. *Microbiology (Reading)* 155:2845-2855.
18. Nicoletti I, Migliorati G, Pagliacci MC, Grignani F, Riccardi C. 1991. A rapid and simple method for measuring thymocyte apoptosis by propidium iodide staining and flow cytometry. *J Immunol Methods* 139:271-9.
19. Hurdle JG, O'Neill AJ, Chopra I, Lee RE. 2011. Targeting bacterial membrane

function: an underexploited mechanism for treating persistent infections. *Nat Rev Microbiol* 9:62-75.

20. Stratford JP, Edwards CLA, Ghanshyam MJ, Malyshev D, Delise MA, Hayashi Y, Asally M. 2019. Electrically induced bacterial membrane-potential dynamics correspond to cellular proliferation capacity. *Proc Natl Acad Sci U S A* 116:9552-9557.

February 2, 2023

Prof. Guocai Li
Yangzhou University Medical college
Pathogen Biology and Immunology
11 Huai-hai Road
Yangzhou, Jiangsu 225001
China

Re: Spectrum04702-22R1 (Baicalein Resensitizes Multidrug-Resistant Gram-negative Pathogens to Doxycycline)

Dear Prof. Guocai Li:

Link Not Available

Sincerely,

Krisztina Papp-Wallace

Journals Department
Reviewer comments:

Reviewer #1 (Comments for the Author):

Thank you for better filling out the methods and legends by including more explanation of how experiments were done. I agree with your decision to retain the mechanistic experiments. While they are not necessarily conclusive with regard to baicalein mechanism of action, we need to understand how bacteria and metazoans respond to potential antimicrobials and adjuvants to develop new approaches to therapy.

Reviewer #2 (Comments for the Author):

Since the original submission, several experiments have been repeated, and the manuscript has been reorganized and edited. Most of the concerns described in the previous reviews have been addressed, at least in part. Some specific comments follow.

1. Mechanistic studies. In responses to concerns by both previous reviewers, most of these experiments have been repeated with much shorter incubation times. In some cases, added controls were added (polymyxin or colistin to propidium iodide and NPN staining; baicalein-only to transcriptomics). It would have been preferable to include antibiotic controls also in other assays, such as ATP, ROS, NADH, PMF, etc. However, the shorter incubation times make these studies much more interpretable and less likely to simply reflect reduced bacterial growth - as would happen with any effective antibiotic.

2. Animal infection. The lung infection study is nice and should be included. I continue to believe that the Galleria and mouse peritonitis studies have little value and should be omitted. My reasoning is as follows. The goal of a mouse efficacy study is to see whether the drug can travel through the bloodstream to the site of infection and have a beneficial physiologic effect. If the drug is administered directly to the site of infection (both injected IP), very little information is obtained. Almost any compound (or combination) with an MIC will appear to be efficacious. Thus the IP/IP model is not helpful in determining whether a compound justifies further effort. Pharmaceutical antibiotic-discovery researchers usually prefer animal models with a quantitative readout (such as cfu in lung or another organ). See examples below of papers describing early animal studies for several antibacterial compounds that were found to be effective for treating human infections. Note that these include experiments monitoring survival following IP infection, but in these cases the drug was administered either IV or subcutaneously.

The authors' response letter provided nine references, from five or so academic research groups around the world, illustrating that this use of mice as "furry test tubes" is a common error.

With regard to Galleria: Invertebrate infection models are a useful alternative to in-vitro susceptibility testing if you are seeking inhibitors of virulence mechanisms that may not be essential in standard broth media. These models might also be useful as an alternative to mice if compound supply is extremely limited, or if you are testing so many compounds that mouse testing is not feasible.

3. Serum - The previous review asked about the effect of serum on antibacterial activity. The response letter indicated the serum in cell culture medium had been heat-inactivated. This misses the point. The question is whether baicalein binds to serum proteins. It is very common to think a compound lacks cytotoxicity, only to find that the compound has been sequestered by serum in the culture medium. The simplest way to evaluate this possibility is to add serum to the bacterial culture medium and see if the MIC is shifted upward.

4. Methods. Suggest careful re-reading of this section to be sure that it accurately reflects the procedures. Line 314, "bacterium solution" should be "bacterial suspension" Line 315, did you actually culture supernatants (after centrifugation), or did you take aliquots of the bacterial culture for dilution and plating?

5. Check for inconsistencies. One example noted: Time-kill curves are described as indicating bacteriostatic activity on line 104, but bactericidal on line 756. Also note the checkerboard MIC assay is described (line 97) as showing bacteriostatic activity. Suggest referring here to antibacterial activity, as an MIC determination cannot distinguish static from cidal activity.

6. The English is improved, but there are still some badly written sentences. Examples:

177-178: "We found that the combination treatment sharply fell cell membrane fluidity". (Maybe: We found that the combination treatment resulted in a sharp fall in cell membrane fluidity.)

246-247: "Baicalein altered bacterial metabolism and decreased the multidrug efflux pump to potentiate the antibacterial activity of doxycycline" (should be "...decreased multidrug efflux pump activity...")

277-278: "Our finding of baicalein can..." (maybe: "We found that baicalein can...")

280: "which causes it challenging..." (maybe: "which makes it challenging...")

--- References for pharmacology:

- Grossman TH et al. 2015. Eravacycline (TP-434) Is Efficacious in Animal Models of Infection. *Antimicrob Agents Chemother* 59:2567-2571. doi:10.1128/AAC.04354-14.

- Macone AB et al. 2014. In vitro and in vivo antibacterial activities of omadacycline, a novel aminomethylcycline. *Antimicrob Agents Chemother* 58:1127-1135. doi:10.1128/AAC.01242-13.

- Durand-Reville TF et al. 2021. Rational design of a new antibiotic class for drug-resistant infections. *Nature* 1-5. doi:10.1038/s41586-021-03899-0.

- Durand-Réville TF et al. 2017. ETX2514 is a broad-spectrum β -lactamase inhibitor for the treatment of drug-resistant Gram-negative bacteria including *Acinetobacter baumannii*. *Nat Microbiol* 2:nmicrobiol2017104. doi:10.1038/nmicrobiol.2017.104.

Staff Comments:

Preparing Revision Guidelines

Please return the manuscript within 60 days; if you cannot complete the modification within this time period, please contact me. If you do not wish to modify the manuscript and prefer to submit it to another journal, please notify me of your decision immediately so that the manuscript may be formally withdrawn from consideration by Microbiology Spectrum.

Comments of Reviewer 1

General comments: Thank you for better filling out the methods and legends by including more explanation of how experiments were done. I agree with your decision to retain the mechanistic experiments. While they are not necessarily conclusive with regard to baicalein mechanism of action, we need to understand how bacteria and metazoans respond to potential antimicrobials and adjuvants to develop new approaches to therapy.

Overall response to Reviewer 1: We are grateful for your recognition of the value of this study. In the revised version, the new text is highlighted in yellow, and the underline indicates the revised text to identify better the changes made to the previous version.

In order to provide a more visual description about how baicalein resensitizes *tetA*-positive MDR gram-negative pathogens to doxycycline, we have added a figure illustrating the “(Figure 8G) The mechanism of action of baicalein combined with doxycycline against *tetA*-positive gram-negative bacteria. Baicalein can restore the susceptibility of doxycycline to MDR gram-negative bacteria via membrane-mediated processes by interacting with LPS in the outer membrane and targeting phospholipids in the cytoplasmic membrane. These processes cause membrane depolarization and malfunction, which in turn leads to metabolic alterations such as an increase in the TCA cycle activity and oxidative damage. Doxycycline can also bind to bacterial ribosomes and inhibit the production of bacterial proteins such as *tetA*.” (Page 46, line 876-884)

In vivo studies were conducted using a neutropenic mouse lung infection model, we found that “Following the addition of baicalein, the bacterial loads in the

bronchoalveolar lavage fluid and mouse organs were the lowest in the combination treatment group (Figure 9). Additionally, the levels of four inflammatory cytokines (IL-1 β , IFN- γ , TNF- α , and IL-8) in serum were considerably lower after 48 h of pulmonary infection in the combination treatment group than that in other groups, which was confirmed by the alleviated pulmonary pathology (Figure S7). These results highlighted that baicalein combined with doxycycline can inhibit bacterial invasion into host cells and reduce the host inflammatory response to these infections *in vivo*.” (Page 11, line 216-224)

Comments of Reviewer 2

General comments: *Since the original submission, several experiments have been repeated, and the manuscript has been reorganized and edited. Most of the concerns described in the previous reviews have been addressed, at least in part. Some specific comments follow.*

Overall response to Reviewer 2: Thank you for reviewing the manuscript and for your positive and constructive comments. In the revised version, the new text is highlighted in yellow, and the underline indicates the revised text to identify better the changes made to the previous version. Our point-to-point response is provided below. We hope these changes improve the clarity and accuracy of the presentation.

In what follows, we would like to answer the questions you mentioned and give a detailed account of the changes made to the original manuscript.

Comment 1: *Mechanistic studies. In responses to concerns by both previous reviewers, most of these experiments have been repeated with much shorter incubation times. In some cases, added controls were added (polymyxin or colistin to propidium iodide and NPN staining; baicalein-only to transcriptomics). It would have been preferable to include antibiotic controls also in other assays, such as ATP, ROS, NADH, PMF, etc. However, the shorter incubation times make these studies much more interpretable and less likely to simply reflect reduced bacterial growth - as would happen with any effective antibiotic.*

Response: It is really a good idea as reviewer suggested, and we have changed them all to meet reviewer's suggestion, we have added antibiotic controls in other assays, such as polymyxins B (16 µg/mL) was added as antibiotic control in extracellular β-galactosidase measurement (Figure 3B; supplementary Figure 3B), ATP content

detection (Figure 3C; supplementary Figure 3C and supplementary Figure 6F), total ROS, superoxide dismutase (SOD), hydrogen peroxide (H₂O₂) measurement (supplementary Figure 5B, 5C and 5E) and biofilm formation analysis (supplementary Figure 6G).

As an antibiotic control, colistin (4 µg/mL) was added into the TCA cycle measurement (supplementary Figure 5A), membrane fluidity assay (supplementary Figure 6A), and PMF (supplementary Figure 6B).

Additionally, we have added the viable counts and bacterial live/dead staining “Relative to the doxycycline alone, 62.5 µg/mL baicalein exhibited significantly enhanced antimicrobial activity of doxycycline and reduced the number of viable bacteria against AB145 and KP1 after 16 h of treatment in vitro (Figure 2C). The results of the live/dead bacteria staining assay showed that the cell survival rate in the combined drug group was lower than that in the single-agent group (Figure 2D-E).”

(Page 6, line 106-110)

Comment 2: Animal infection. The lung infection study is nice and should be included. I continue to believe that the Galleria and mouse peritonitis studies have little value and should be omitted. My reasoning is as follows. The goal of a mouse efficacy study is to see whether the drug can travel through the bloodstream to the site of infection and have a beneficial physiologic effect. If the drug is administered directly to the site of infection (both injected IP), very little information is obtained. Almost any compound (or combination) with an MIC will appear to be efficacious. Thus the IP/IP model is not helpful in determining whether a compound justifies further effort. Pharmaceutical antibiotic-discovery researchers usually prefer animal models with a quantitative readout (such as cfu in lung or another organ). See examples below of

papers describing early animal studies for several antibacterial compounds that were found to be effective for treating human infections. Note that these include experiments monitoring survival following IP infection, but in these cases the drug was administered either IV or subcutaneously.

The authors' response letter provided nine references, from five or so academic research groups around the world, illustrating that this use of mice as "furry test tubes" is a common error.

With regard to Galleria: Invertebrate infection models are a useful alternative to in-vitro susceptibility testing if you are seeking inhibitors of virulence mechanisms that may not be essential in standard broth media. These models might also be useful as an alternative to mice if compound supply is extremely limited, or if you are testing so many compounds that mouse testing is not feasible.

Response: Thank you for the comment, with which we fully agree. We have removed the *Galleria mellonella* and mouse peritonitis studies, and revised the text accordingly.

“Baicalein improves doxycycline efficacy in vivo. Baicalein combined with doxycycline showed an effective antibacterial activity in vitro (Figure 8G), which prompted further analysis of this combinatorial treatment in infection models. We used a neutropenic mouse lung infection model to evaluate the effects of this combinatorial treatment in vivo. Following the addition of baicalein, the bacterial loads in the bronchoalveolar lavage fluid and mouse organs were the lowest in the combination treatment group (Figure 9). Additionally, the levels of four inflammatory cytokines (IL-1 β , IFN- γ , TNF- α , and IL-8) in serum were considerably lower after 48 h of pulmonary infection in the combination treatment group than that in other groups, which was confirmed by the alleviated pulmonary pathology (Figure S7). These

results highlighted that baicalein combined with doxycycline can inhibit bacterial invasion into host cells and reduce the host inflammatory response to these infections in vivo." (Page 10, line 212-224).

Comment 3: Serum - The previous review asked about the effect of serum on antibacterial activity. The response letter indicated the the serum in cell culture medium had been heat-inactivated. This misses the point. The question is whether baicalein binds to serum proteins. It is very common to think a compound lacks cytotoxicity, only to find that the compound has been sequestered by serum in the culture medium. The simplest way to evaluate this possibility is to add serum to the bacterial culture medium and see if the MIC is shifted upward.

Response: We appreciate the reviewer's insightful suggestion and have updated. To exclude the possibility of a protective effect of serum in the culture medium against agent-induced cytotoxicity, 10% fetal bovine serum (FBS) and combinations of drugs were also added to the checkerboard assays. The results showed that while the MIC for baicalein and doxycycline did not change, and these two drugs still had synergistic functions, but the FIC of AB145 (from 0.09375 increased to 0.1875) and KP1 (from 0.1875 increased to 0.375) were all elevated. In order to thoroughly rule out the protective effects of serum against drugs and toxic agents, serum-free medium was used during drug treatment period. "When the cells reached 80% to 90% confluence, the medium was discarded and the cells were washed by PBS three times and cultured in the corresponding serum-free medium (1-3). Doxycycline (0–128 µg/mL) in combination with baicalein (2–250 µg/mL) was added to 96-well plates along with 1×10^4 cells, which were cultured for 24 h at 37°C. After 24 h of culture, the CCK-8 solution (10 µL/well) was added to each well. The OD values were determined at 450 nm to evaluate the cytotoxic effect of doxycycline combined with baicalein." (Page

16, line 344-350).

The results are illustrated in supplementary Figure 1. All cells showed a survival rate of more than 80%. According to the International Organization for Standardization (ISO), a substance is defined as cytotoxic if the cell survival rate is less than 70% (4). “The mammalian cell cytotoxicity results consistently showed that the synergism of baicalein and doxycycline had low cytotoxicity (Figure S1B-D) (4).” (Page 6, line 114-116).

Comment 4: Methods. Suggest careful re-reading of this section to be sure that it accurately reflects the procedures. Line 314, "bacterium solution" should be "bacterial suspension" Line 315, did you actually culture supernatants (after centrifugation), or did you take aliquots of the bacterial culture for dilution and plating?

Response: We feel sorry for the inconvenience brought to the reviewer. Regarding the suggestion about the methods, we revised and modified this section to ensure that it accurately reflects the procedures.

We’ve changed “All strains were cultivated using Mueller Hinton broth (MHB; Oxoid Ltd., Cambridge, UK) at 37°C and 220 rotations per minute (rpm) unless stated otherwise.” to “All strains were cultivated in Mueller Hinton broth (MHB; Oxoid Ltd., Cambridge, UK) at 37°C and 220 rotations per minute (rpm) unless stated otherwise.” (Page 14, line 298-299).

“RAW264.7 and A549 cells were passaged in DMEM (Gibco) with 10% heat-inactivated FBS (Invitrogen).” to “RAW264.7 and A549 cells were cultured in DMEM (Gibco) with 10% heat-inactivated FBS (Invitrogen).” (Page 14, line 302-303).

“Tested antibiotics were added in triplicate into flat-bottomed 96-well plate (Corning) wells, followed by bacterial inoculum supplementation (0.5×10^6 CFUs/mL).” to “Tested antibiotics were added into triplicate wells of 96-well flat-bottomed tissue culture plates (Corning), and two-fold serial dilution was performed, followed by the addition of the prepared bacterial inoculum (0.5×10^6 CFUs/mL).” (Page 14, line 310-312).

“When required, HCl or NaOH was supplemented to the medium to adjust the pH to 5-9.” to “HCl or NaOH was added to the medium to adjust the pH to 5–9.” (Page 15, line 314-315).

“**Time-kill curve.** An overnight culture was diluted with sterile MHB at 1:100 and incubated for 4 hours (exponential phase) at 37°C with shaking at 220 rpm. Then sub-MIC of baicalein (125 µg/mL) and doxycycline (64 µg/mL) was added to the bacterium solution separately or combinatorially incubated for 24 hours. At the corresponding time, different treated cocultured supernatants were collected and serially 10-fold diluted,” to “**Bacterial growth inhibition assay.** An overnight culture was diluted with sterile MHB at 1:100 and incubated for 4 h (exponential phase) at 37°C with shaking at 220 rpm. Then, sub-MIC of baicalein (125 µg/mL) and doxycycline (64 µg/mL) were added to the bacterial suspension separately or in combination and incubated for 24 h. At the corresponding time, differently treated co-cultured aliquots of the bacterial culture were collected and serially 10-fold diluted” (Page 15, line 321-326).

“**Resistance development studies.** Bacterium solution (0.5×10^6 CFUs/mL) contains doxycycline ($0.25 \times$ MIC) alone or in combination with baicalein (62.5 µg/mL) incubated at 220 rpm for 24 hours (37°C). Subsequently, the visible growth ($OD_{600} \geq$

0.3) strains were passaged on antibiotic-free MHA plates for 24 h, and the corresponding MICs of doxycycline were detected by the two-fold serial dilution method described above. Simultaneously, the bacterium solution ($OD_{600} = 0.5$) was diluted 1:100 with MHB and incubated in the presence of doxycycline (1/4 MIC) or combination with baicalein (62.5 $\mu\text{g}/\text{mL}$) for the next generation. This step was performed repeatedly for 30 days; then, the fold increase of the doxycycline MIC compared with the original MIC was determined” to “**Resistance development studies.** Bacterial cultures (0.5×10^6 CFUs/mL) containing doxycycline ($0.25 \times \text{MIC}$) alone or in combination with baicalein (62.5 $\mu\text{g}/\text{mL}$) were incubated at 220 rpm for 24 h (37°C). Subsequently, the visible growth ($OD_{600} \geq 0.3$) strains were passaged on antibiotic-free MHA plates for 24 h, and the corresponding MICs of doxycycline were detected by the two-fold serial dilution method described above. The bacterial suspension ($OD_{600} = 0.5$) was diluted 1:100 with MHB and incubated in the presence of doxycycline (1/4 MIC) alone or in combination with baicalein (62.5 $\mu\text{g}/\text{mL}$) for the next generation. This step was performed repeatedly for 30 days” (Page 16, line 351-359).

“Different concentrations of baicalein or PB (polymyxins B, 16 $\mu\text{g}/\text{mL}$) were co-incubated with bacterium solution ($OD_{600} = 0.5$) for 30 mins (37°C).” to “Different concentrations of baicalein or PB (polymyxins B, 16 $\mu\text{g}/\text{mL}$) were co-incubated with bacterial suspension ($OD_{600} = 0.5$) for 30 mins (37°C).” (Page 17, line 377-379).

“**Measurement of ATP content.** Bacterium solution ($OD_{600} = 0.5$) was co-incubated with the indicated concentrations of drugs for 30 mins.” to “**Measurement of ATP content.** The bacterial suspension ($OD_{600} = 0.5$) was co-incubated with the indicated concentrations of drugs for 30 min.” (Page 17, line 382-383).

“Scanning electron microscopy (SEM). Doxycycline (64 $\mu\text{g/mL}$), baicalein (125 $\mu\text{g/mL}$), or their combination was used to treat bacterium solution ($\text{OD}_{600} = 0.5$) for 1 h, followed by fixed overnight using 2.5% glutaraldehyde at 4°C. After thoroughly rinsing thrice using PBS, bacteria were subjected to 10 min gradual dehydration using ethanol (30%, 50%, 70%, 80%, 90%, 95%) and 10 min dehydration using 100% ethanol thrice. The samples were centrifuged at 4,000 g for 10 min to remove fixatives, followed by resuspension of bacterial pellets in PBS (1 mL). After processing, each sample was dried using a critical point dryer, coated with gold–palladium using an ion sprayer” to **“Scanning electron microscopy (SEM).** Doxycycline (64 $\mu\text{g/mL}$), baicalein (125 $\mu\text{g/mL}$), or their combination was used to treat bacterial suspension ($\text{OD}_{600} = 0.5$) for 1 h, followed by overnight incubation with 2.5% glutaraldehyde at 4°C. After thoroughly rinsing thrice using PBS, the bacteria were subjected to gradual dehydration in the sequence of 30%, 50%, 70%, 80%, 90%, and 95% ethanol for 10 min each and 100% ethanol for 10 min thrice. Subsequently, the samples were dried in a critical point dryer, coated with gold-palladium using an ion sprayer” (Page 18, line 387-393).

“The checkerboard microdilution assay was applied to assess how LPS (0-128 $\mu\text{g/mL}$) affected baicalein's antibacterial effect. MIC assays were used to determine the impact of different cationic ions (50 $\mu\text{g/mL}$), which included KCl, CaCl₂, MgCl₂, CuSO₄, FeCl₃, NaCl, MgCl₂, and ZnSO₄ (Aladdin, Shanghai, China), on baicalein's antibacterial effect on *A. baumannii* or *K. pneumoniae*. Phosphatidylglycerol (PG; Sigma-Aldrich, 841188P, $\geq 99\%$), phosphatidylcholine (PC; Sigma-Aldrich, catalog no. 840051P, $\geq 99\%$), cardiolipin (CL; Sigma-Aldrich, 841199P, $\geq 99\%$), and phosphatidylethanolamine (PE; Sigma-Aldrich, 840027P, $\geq 99\%$) were subjected to methanol dissolution. The checkerboard microdilution assay was used to assess how different phospholipid (0–12

8 µg/mL) concentrations affected baicalein's antibacterial effect in MHB.” to “The checkerboard microdilution assay was performed to assess how LPS (0-128 µg/mL) affected the antibacterial activity of baicalein. MIC assays were performed to determine the impact of different cationic ions (50 µg/mL), including KCl, CaCl₂, MgCl₂, CuSO₄, FeCl₃, NaCl, MnSO₄, and ZnSO₄ (Aladdin, Shanghai, China), on the antibacterial activity of baicalein on *A. baumannii* or *K. pneumoniae*. Phosphatidylglycerol (PG; Sigma-Aldrich, 841188P, ≥99%), phosphatidylcholine (PC; Sigma-Aldrich, catalog no. 840051P, ≥99%), cardiolipin (CL; Sigma-Aldrich, 841199P, ≥99%), and phosphatidylethanolamine (PE; Sigma-Aldrich, 840027P, ≥99%) were dissolved into methanol. The checkerboard microdilution assay was performed to determine the effects of different phospholipid concentrations (0–128 µg/mL) on the antimicrobial activity of baicalein in MHB.” (Page 18, line 398-408).

“**RT-PCR and transcriptomic analysis.** Bacterial cells with an OD₆₀₀ of 0.5 were treated with the indicated concentrations of baicalein at 37°C for 30 mins.” to “**RT-PCR and transcriptomic analysis.** Bacterial suspension with an OD₆₀₀ of 0.5 was treated with the indicated concentrations of baicalein at 37°C for 30 min.” (Page 19, line 409-410).

“Next, the bacterial solution was mixed with varying concentrations of baicalein, baicalein combined with doxycycline, or benzyl alcohol (50 mmol/L) for 30 mins.” to “Next, the bacterial culture was mixed with varying concentrations of baicalein, baicalein combined with doxycycline, colistin (4 µg/mL), or benzyl alcohol (50 mmol/L) for 30 min.” (Page 20, line 446-448).

“**PMF and membrane depolarization assay.** Bacterial cells with an OD₆₀₀ of 0.5 co-incubated with BCECF-AM” to “**PMF and membrane depolarization assay.**

Bacterial suspension with an OD₆₀₀ of 0.5 was co-incubated with BCECF-AM (0.2 × 10⁻⁶ M)” (Page 20, line 452-453).

“**Ethidium bromide (EtBr) efflux and biofilm formation analysis.** Bacterium solution (OD₆₀₀ = 0.5), EtBr (5 × 10⁻⁶ M), the indicated concentrations of drugs” to “**Ethidium bromide (EtBr) efflux and biofilm formation analysis.** The bacterial suspension (OD₆₀₀ = 0.5), EtBr (5 × 10⁻⁶ M), drugs at specific concentrations” (Page 21, line 459-460).

“Mouse organs were dissected under septic conditions, followed by homogenization, serial dilution, and inoculation in MHA for CFU counting. Meanwhile, those mice's left upper lung tissues were excised aseptically for hematoxylin and eosin (HE) staining.” to “The organs of the mice were dissected under sterile conditions, followed by homogenization, serial dilution, and inoculation in MHA for counting CFUs. The left upper lung tissues of these mice were excised aseptically for hematoxylin and eosin (HE) staining.” (Page 22, line 489-492).

In addition, some experimental procedures and reagents were added in the Methods section. “The viability of the bacteria was determined by the CFUs/mL after incubation for 16 h (5).” (Page 15, line 327-329).

“**Bacterial live/dead staining.** Bacteria cultured in the logarithmic phase of growth were diluted to 1 × 10⁶ CFU/mL in MHB. The bacterial suspensions were incubated with baicalein (62.5 μg/mL) or doxycycline (32 μg/mL) alone or in combination for 8 h. The cells were then harvested by centrifugation and resuspended in sterile PBS. The LIVE/DEAD BacLight bacterial viability kits (Molecular Probes) containing SYTO9 and PI were used to distinguish between live and dead bacteria. The samples were visualized using a Zeiss Axio Scope A1 upright microscope, and the images

were processed to merge using the ImageJ software (6).” (Page 15, line 330-337).

Comment 5: Check for inconsistencies. One example noted: Time-kill curves are described as indicating bacteriostatic activity on line 104, but bactericidal on line 756. Also note the checkerboard MIC assay is described (line 97) as showing bacteriostatic activity. Suggest referring here to antibacterial activity, as an MIC determination cannot distinguish static from cidal activity.

Response: Thank you so much for your careful check. We have made correction according to the reviewer’s comments. We’ve revised “a checkerboard assay showed that baicalein could act as a synergistic bacteriostatic effect ($FICI \leq 0.5$)” to “a checkerboard assay showed that baicalein can act as a synergistic antibacterial activity ($FICI \leq 0.5$)” (Page 6, line 96-98).

“Our time-dependent killing curve showed that the combination treatment had good bacteriostatic effects on the *tetA*-positive MDR *A. baumannii* AB145 ($FICI = 0.09375$)” to “Our time-dependent killing curve showed that the combination treatment had good antibacterial activity on the *tetA*-positive MDR *A. baumannii* AB145 ($FICI = 0.09375$)” (Page 6, line 103-105).

“and high concentrations of LPS (128 $\mu\text{g/mL}$) diminished the antibacterial effects of baicalein combined doxycycline” to “and high concentrations of LPS (128 $\mu\text{g/mL}$) reduced the antibacterial activity of baicalein combined with doxycycline” (Page 7, line 141-143).

“and Mg^{2+} strongly inhibited the antibacterial effects of baicalein combined doxycycline” to “and Mg^{2+} strongly inhibited the antibacterial activity of baicalein combined with doxycycline” (Page 8, line 146-148).

“The MICs of baicalein are higher with the exogenous addition of phospholipids, and

PG blocks the antibacterial effects of baicalein combined with antibiotics” to “The MICs of baicalein were higher with the exogenous addition of phospholipids, and PG blocked the antibacterial activity of baicalein combined with antibiotics” (Page 8, line 154-156).

“**Transcriptomic analysis of the antibacterial effects of baicalein combined with doxycycline.**” to “**Transcriptomic analysis of the antibacterial activity of baicalein combined with doxycycline.**” (Page 8, line 159-160).

“Thus, the antibacterial effects of baicalein combined with doxycycline may be associated with multiple pathways.” to “Thus, the antibacterial activity of baicalein combined with doxycycline might be associated with multiple pathways.” (Page 9, line 173-174).

“Baicalein combined with doxycycline had a practical antibacterial effect in vitro” to “Baicalein combined with doxycycline showed an effective antibacterial activity in vitro” (Page 10, line 212-213).

“Baicalein exhibited synergistic antibacterial effects by disrupting gram-negative bacteria's inner and outer membranes.” to “Baicalein exhibited synergistic antibacterial activity by disrupting the inner and outer membranes of gram-negative bacteria.” (Page 12, line 240-241).

“**Baicalein synergy with doxycycline has a bactericidal effect on *tetA*-positive gram-negative pathogens in vitro.**” to “**Baicalein synergy with doxycycline has an antibacterial activity on *tetA*-positive gram-negative pathogens in vitro.**” (Page 37, line 769-770).

“**Figure 3. Baicalein exerts antibacterial effects through the inner and outer**

membranes of *A. baumannii*.” to “**Figure 3. Baicalein exerts antibacterial activity through the inner and outer membranes of *A. baumannii*.**” (Page 39, line 792-793).

“(B) Purified LPS (128 µg/mL) reduces the antibacterial effects of baicalein combined with doxycycline.” to “(B) Purified LPS (128 µg/mL) reduced the antibacterial activity of baicalein combined with doxycycline.” (Page 41, line 811-812).

“(E) The exogenous addition of Mg²⁺ (480 µg/mL) reduces the antibacterial effects of baicalein combined with doxycycline.” to “(E) The exogenous addition of Mg²⁺ (480 µg/mL) reduced the antibacterial activity of baicalein combined with doxycycline.” (Page 41, line 816-817).

“(C) PG blocks the antibacterial effects of baicalein combined with multiple antibiotics” to “(C) PG blocks the antibacterial activity of baicalein combined with multiple antibiotics” (Page 42, line 829-830).

Comment 6: *The English is improved, but there are still some badly written sentences.*

Examples: 177-178: "We found that the combination treatment sharply fell cell membrane fluidity". (Maybe: We found that the combination treatment resulted in a sharp fall in cell membrane fluidity.)

246-247: "Baicalein altered bacterial metabolism and decreased the multidrug efflux pump to potentiate the antibacterial activity of doxycycline" (should be "...decreased multidrug efflux pump activity...")

277-278: "Our finding of baicalein can..." (maybe: "We found that baicalein can...")

280: "which causes it challenging..." (maybe: "which makes it challenging...")

Response: Thanks for your constructive suggestion, which is highly appreciated. We have carefully scrutinized the manuscript, and made corresponding revisions including some typos, grammatical errors and long sentences, ect. In addition, the

manuscript has been polished by an English language editing company. The certificate of English language editing has been uploaded to the system along with the revised manuscript.

We tried our best to improve the manuscript and made some changes in the manuscript. These changes will not influence the content and framework of the paper. And here, we did not list the changes but marked them in red in the revised paper.

We appreciate for Editors/Reviewers' warm work earnestly and hope that the correction will meet with approval.

We look forward to hearing from you regarding our submission and responding to any further questions and comments you may have.

Once again, thank you very much for your comments and suggestions.

Sincerely,

Corresponding author:

Name: Guocai Li

E-mail: gcli@yzu.edu.cn

References

1. Gauthier AG, Lin M, Zefi S, Kulkarni A, Thakur GA, Ashby CR, Jr., Mantell LL. 2023. GAT107-mediated $\alpha 7$ nicotinic acetylcholine receptor signaling

- attenuates inflammatory lung injury and mortality in a mouse model of ventilator-associated pneumonia by alleviating macrophage mitochondrial oxidative stress via reducing MnSOD-S-glutathionylation. *Redox Biol* 60:102614.doi.org/10.1016/j.redox.2023.102614
2. Tóth E, Turiák L, Visnovitz T, Cserép C, Mázló A, Sódar BW, Försönits AI, Petővári G, Sebestyén A, Komlósi Z, Drahos L, Kittel Á, Nagy G, Bácsi A, Dénes Á, Ghó YS, Szabó-Taylor K, Buzás EI. 2021. Formation of a protein corona on the surface of extracellular vesicles in blood plasma. *J Extracell Vesicles* 10:e12140.doi.org/10.1002/jev2.12140
 3. Wang Q, Fang P, He R, Li M, Yu H, Zhou L, Yi Y, Wang F, Rong Y, Zhang Y, Chen A, Peng N, Lin Y, Lu M, Zhu Y, Peng G, Rao L, Liu S. 2020. O-GlcNAc transferase promotes influenza A virus-induced cytokine storm by targeting interferon regulatory factor-5. *Sci Adv* 6:eaz7086.doi.org/10.1126/sciadv.aaz7086
 4. ISO 10993–5:2009(E), Biological Evaluation of Medical Devices-Part 5: Tests for in Vitro Cytotoxicity.(International Organization for Standardization,2009)
 5. She P, Li S, Zhou L, Luo Z, Liao J, Xu L, Zeng X, Chen T, Liu Y, Wu Y. 2020. Insights into idarubicin antimicrobial activity against methicillin-resistant *Staphylococcus aureus*. *Virulence* 11:636-651.doi.org/10.1080/21505594.2020.1770493
 6. She P, Li Z, Li Y, Liu S, Li L, Yang Y, Zhou L, Wu Y. 2022. Pixantrone Sensitizes Gram-Negative Pathogens to Rifampin. *Microbiol Spectr* 10:e0211422.doi.org/10.1128/spectrum.02114-22

March 31, 2023

Prof. Guocai Li
Yangzhou University Medical college
Pathogen Biology and Immunology
11 Huai-hai Road
Yangzhou, Jiangsu 225001
China

Re: Spectrum04702-22R2 (Baicalein Resensitizes Multidrug-Resistant Gram-negative Pathogens to Doxycycline)

Dear Prof. Guocai Li:

Your manuscript has been accepted, and I am forwarding it to the ASM Journals Department for publication. You will be notified when your proofs are ready to be viewed.

Sincerely,

Krisztina Papp-Wallace
Editor, Microbiology Spectrum
